# Multi-functional roles for the polypeptide transport associated domains of Toc75 in chloroplast protein import

Yamuna D Paila[1], Lynn GL Richardson[1], Hitoshi Inoue[1], Elizabeth S Parks[1], James McMahon[1], Kentaro Inoue[2], Danny J Schnell[1]*

[1]Department of Plant Biology, Michigan State University, East Lansing, United States; [2]Department of Plant Sciences, University of California, Davis, United States

**Abstract** Toc75 plays a central role in chloroplast biogenesis in plants as the membrane channel of the protein import translocon at the outer envelope of chloroplasts (TOC). Toc75 is a member of the Omp85 family of bacterial and organellar membrane insertases, characterized by N-terminal POTRA (polypeptide-transport associated) domains and C-terminal membrane-integrated $\beta$-barrels. We demonstrate that the Toc75 POTRA domains are essential for protein import and contribute to interactions with TOC receptors, thereby coupling preprotein recognition at the chloroplast surface with membrane translocation. The POTRA domains also interact with preproteins and mediate the recruitment of molecular chaperones in the intermembrane space to facilitate membrane transport. Our studies are consistent with the multi-functional roles of POTRA domains observed in other Omp85 family members and demonstrate that the domains of Toc75 have evolved unique properties specific to the acquisition of protein import during endosymbiotic evolution of the TOC system in plastids.

*For correspondence: dschnell@cns.msu.edu

**Competing interests:** The authors declare that no competing interests exist.

## Introduction

The biogenesis and function of plastids are reliant on the coordinate expression and selective import of >2500 different nucleus-encoded proteins (*Ling and Jarvis, 2015*, *Shi and Theg, 2013*). The major pathway for protein import involves the synchronized action of two multi-protein membrane translocons, the TOC (translocon at the outer envelope membrane of chloroplasts) and the TIC (translocon at the inner envelope membrane of chloroplasts) complexes (*Paila et al., 2015*, *Lee et al., 2014*), which reside within the chloroplast envelope. TOC and TIC complexes contain selective protein conducting channels, which align via physical association between the complexes to provide simultaneous membrane translocation of preproteins from the cytoplasm to the plastid interior (*Paila et al., 2015*, *Shi and Theg, 2013*, *Ling and Jarvis, 2015*). The selectivity and directionality of protein import is controlled by sequential interactions between the targeting determinants within the cleavable, N-terminal transit peptides of preproteins and binding sites within the TOC and TIC complexes (*Li and Teng, 2013*), as well as import-associated molecular chaperones (*Flores-Pérez and Jarvis, 2013*, *Paila et al., 2015*, *Schwenkert et al., 2011*).

The evolution of the TOC and TIC system during endosymbiosis required the adaptation or acquisition of protein conducting channels, receptors and molecular chaperones at both envelope membranes to provide a unidirectional conduit for the transport of nuclear encoded polypeptides into the organelle. Protein transport across the outer envelope membrane is mediated by a β-barrel membrane protein, Toc75, which is proposed to constitute the major component of the TOC import channel (*Hinnah et al., 2002*). Toc75 assembles with two integral membrane receptor GTPases, Toc159 and Toc34, which recognize plastid preproteins in the cytoplasm and control access of

**eLife digest** Chloroplasts are a hallmark feature of plant cells and the sites of photosynthesis – the process in which plants harness the energy in sunlight for their own needs. The first chloroplasts arose when a photosynthetic bacterium was engulfed by another host cell, and most of the original bacterial genes have been transferred to the host cell's nucleus during the evolution of land plants. As a result, modern chloroplasts need to import the thousands of proteins encoded by these genes from the rest of the cell.

The chloroplast protein import system relies on a protein transporter in the chloroplast membrane that evolved from a family of bacterial transporters. However, the bacterial transporters were initially involved in protein export, and it was not known how the activity of these transporters adapted to move proteins in the opposite direction.

Paila et al. set out to better understand the chloroplast protein import system and produced mutated forms of the transporter in the model plant *Arabidopsis thaliana*. These experiments revealed that a part of the transporter that is conserved in many other organisms, the "protein transport associated domains", has been adapted for three key roles in protein import. First, this part of the transporter interacts with the other components of the import system that make the transporter more selective and control which direction the proteins are transported. Second, the domains interact with proteins during transport to help move them across the chloroplast membrane. Finally, the domains recruit other molecules called chaperones, which stop the protein from aggregating or misfolding during the transport process. These activities are similar to those for the bacterial export transporters, but clearly evolved to allow transport in the opposite direction – that is, to import proteins into chloroplasts.

The next challenges are to explain how proteins destined for chloroplasts are recognized and transported through the chloroplast's membrane.

preproteins to the Toc75 channel via their intrinsic GTPase activities (*Chang et al., 2012*, *Kessler and Schnell, 2009*, *Kessler and Schnell, 2004*). In addition to its role as the TOC channel, Toc75 is proposed to participate in the targeting and insertion of membrane-anchored outer membrane proteins, including the TOC GTPases (*Wallas et al., 2003*).

Toc75 is encoded by a single gene in Arabidopsis (*TOC75-III, AT3G46740*) and null mutants are embryo lethal (*Jackson-Constan and Keegstra, 2001*, *Baldwin et al., 2005*, *Hust and Gutensohn, 2006*). Knockdown of *TOC75-III* expression by RNAi (*Huang et al., 2011*) and a hypomorphic *toc75-III* mutant (*Stanga et al., 2009*) both result in pale phenotypes and growth defects, consistent with the role of Toc75 in chloroplast biogenesis. Based on these studies, Toc75 appears to have evolved to perform multiple functions in protein import, in the assembly of TOC complexes, and in the biogenesis of the outer membrane (*Richardson et al., 2014*).

Toc75 belongs to the outer membrane protein of 85 kDa (Omp85) superfamily of β-barrel integral membrane proteins (*Hsu and Inoue, 2009*). Omp85 proteins are exclusively localized to the outer membranes of Gram-negative bacteria, mitochondria and plastids (*Inoue and Potter, 2004*, *Schleiff et al., 2011*, *Voulhoux and Tommassen, 2004*), and perform major roles in protein transport and the insertion and assembly of outer membrane proteins (*Voulhoux and Tommassen, 2004*, *Schleiff et al., 2011*). For example, BamA (β-barrel Assembly Machinery protein A) and Sam50 (Sorting and Assembly Machinery 50 kDa), two members of the Omp85 family, function in outer membrane biogenesis in Gram-negative bacteria and mitochondria, respectively, by facilitating the folding and insertion of β-barrel proteins at the outer membrane from the periplasmic or intermembrane space (*Voulhoux et al., 2003*, *Noinaj et al., 2013*, *Schleiff and Soll, 2005*). The structural similarity of Toc75 to these conserved membrane transporters led to the hypothesis that Toc75 evolved from an ancestral Omp85 gene and was adapted during endosymbiosis of the cyanobacterial ancestor to function as the protein import channel in the plastid outer envelope (*Day et al., 2014*, *Topel et al., 2012*, *Bredemeier et al., 2007*, *Gentle et al., 2005*).

Toc75 exhibits two main structural features characteristic of Omp85 superfamily members. The N-terminal ~30 kDa region of Toc75 contains three repeats of POTRA (polypeptide-transport

associated) domains, each characterized by a $\beta_1\alpha_1\alpha_2\beta_2\beta_3$ secondary structural motif (*Clantin et al., 2007*, *Koenig et al., 2010*, *Paila et al., 2015*). The ~45 kDa C-terminal region is predicted to contain 16 membrane-spanning β-strands and constitute the membrane-integrated β-barrel (*Paila et al., 2015*, *Day et al., 2014*). While there is significant evidence that the C-terminal β-barrel domain of Toc75 acts as a component of the translocon channel for protein translocation at the outer membrane (*Hinnah et al., 1997*), the functions of the N-terminal POTRA domains have not been clearly defined.

In the case of BamA, specific POTRA repeats are indispensable for the insertion of β-barrel proteins at the *E. coli* outer membrane (*Browning et al., 2013*). The POTRA domains of BamA extend into the periplasm, where they interact with other components of the BAM complex (Bam B-E), a major periplasmic chaperone, SurA (*Bennion et al., 2010*), and nascent outer membrane proteins (*Kim et al., 2007*, *Noinaj et al., 2015*). Structural studies of BamA also suggest that specific POTRA repeats interact with the C-terminal domain to act as a possible gate to the membrane channel of the β-barrel (*Noinaj et al., 2013*, *Noinaj et al., 2015*, *Bakelar et al., 2016*). In another example, deletion of a major portion of the POTRA domain of Sam50 in yeast mitochondria inhibits growth (*Habib et al., 2007*), and subsequent studies provided evidence that the domain interacts with β-barrel precursors to promote their release from the SAM complex and insertion in the outer membrane from the intermembrane space (*Kutik et al., 2008*, *Stroud et al., 2011*). The studies of BamA and Sam50 are consistent with models in which the POTRA repeats are assembled into multi-functional cassettes to mediate protein-protein interactions required for the assembly and specific targeting function of the membrane biogenesis machinery (*Koenig et al., 2010*).

The relationship between Toc75 and other Omp85 superfamily members raises interesting evolutionary and mechanistic questions of how the structural and functional features of the conserved POTRA and β-barrel domains of Toc75 have adapted during endosymbiosis to constitute the protein import channel. Structural and electrophysiological studies of Toc75 demonstrate that the protein forms a cation-selective channel with a pore size sufficient to transport unfolded polypeptide substrates (*Hinnah et al., 1997*; *2002*). In vitro pull-down experiments with recombinant POTRA domains of Toc75 suggest that they can interact with the TOC GTPases and chloroplast preproteins (*Ertel et al., 2005*). This has led to the proposal that they function in TOC complex assembly, interactions with the TOC GTPases, preprotein recognition, or to provide a chaperone-like activity for the preprotein during membrane translocation (*Sánchez-Pulido et al., 2003*, *Ertel et al., 2005*). In this study, we combine molecular genetic and biochemical analyses to examine the roles of the POTRA domains in protein import and membrane targeting in chloroplasts. We demonstrate that POTRA domains contribute essential functions to the Toc75 channel. Furthermore, expression of Toc75 lacking one or more of the three POTRA repeats leads to dominant negative phenotypes in *Arabidopsis thaliana*. Expression of specific deletion mutants alters the size and distribution of the TOC complexes and disrupts preprotein import into the organelle. Biochemical analyses indicate that the Toc75 POTRAs interact specifically with chloroplast preproteins and two proposed chaperones of the intermembrane space, Tic22-III and Tic22-IV. Our results demonstrate multiple roles for the POTRAs in protein import, including TOC complex assembly, preprotein translocation and the recruitment of chaperones to facilitate transport across the intermembrane space.

## Results

### POTRA domains are required for Toc75 function

To investigate the role of POTRA domains in the function of Toc75, we introduced a series of in-frame internal deletion mutations in a *TOC75-III* genomic construct to encode proteins lacking one (Toc75ΔP1), two (Toc75ΔP1-2) or all three (Toc75ΔP1-3) predicted POTRA domains (*Figure 1A*), and tested their ability to complement the lethal phenotype of the *toc75-III-1* null mutant. The *TOC75-III* genomic fragment retains both the native promoter and introns and was previously shown to complement *toc75-III-1* (*Shipman-Roston et al., 2010*). Heterozygous *toc75-III-1* plants were transformed with a wild-type TOC75 gene construct (*TOC75*), *TOC75ΔP1*, *TOC75ΔP1-2* or *TOC75ΔP1-3*. Transformants were selected for hygromycin resistance linked to the *toc75-III-1* T-DNA insertion and for the DsRed fluorescence marker linked to the *TOC75* and POTRA-deletion constructs. The presence of *TOC75*, *TOC75ΔP1*, *TOC75ΔP1-2* or *TOC75ΔP1-3* (*Figure 1B*) and the *toc75-III-1* allele

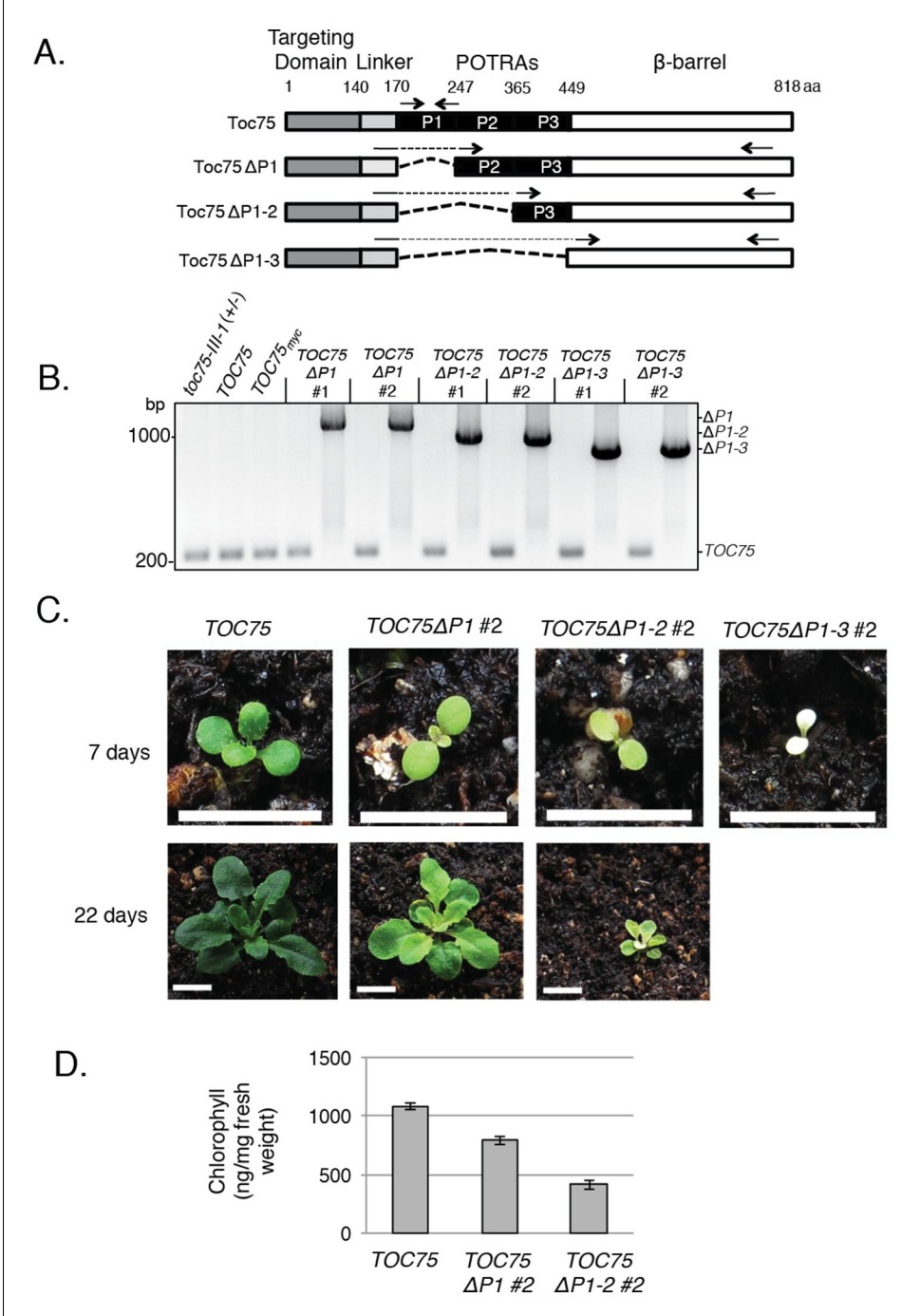

**Figure 1.** Phenotypes of *TOC75ΔP1, TOC75ΔP1-2* or *TOC75ΔP1-3* in *toc75-III-1* plants. (**A**) Schematic diagram of Toc75 and Toc75ΔP1, Toc75ΔP1-2 or Toc75ΔP1-3 proteins used in this study. The dashed line represents the deleted POTRA domain region in various constructs. The numbers refer to the amino acid position, with 1 indicating the N-terminal residue of mature Toc75. (**B**) PCR confirmation of the genotypes of heterozygous *toc75-III-1* plants transformed with *TOC75* and *TOC75ΔP1, TOC75ΔP1-2* or *TOC75ΔP1-3* constructs. Primers spanning the genomic regions encoding POTRA-1, -2 or -3 were used to distinguish between *TOC75* and *TOC75ΔP1, TOC75ΔP1-2* or *TOC75ΔP1-3* (**Table 2**). The approximate positions of the primers are indicated by the forward and reverse arrows. The positions of the PCR products for *TOC75* and *TOC75ΔP1* (ΔP1), *TOC75ΔP1-2* (ΔP1-2) and *TOC75ΔP1-3* (ΔP1-3) are indicated to the right of the figure. (**C**) Visual phenotypes of 7- and 22-day old *TOC75*, *TOC75ΔP1#2, TOC75ΔP1-2#2* and *TOC75ΔP1-3#2* (7-day only) plants grown on soil. Bars = 1 cm (**D**) Chlorophyll content of 22-day old *TOC75, TOC75ΔP1#2, TOC75ΔP1-2#2* plants.

*Figure 1 continued on next page*

*Figure 1 continued*

The following figure supplement is available for figure 1:

**Figure supplement 1.** Phenotypes for individual transgenic *toc75-III-1* lines expressing *TOC75ΔP1* or *TOC75ΔP1-2*.

(data not shown) was confirmed by PCR of genomic DNA in T3 transformants homozygous for the DsRed marker. Plants homozygous for the *TOC75, TOC75ΔP1, TOC75ΔP1-2* or *TOC75ΔP1-3* transgenes were selected in the heterozygous *toc75-III-1* (*toc75III-1*[+/-]) background, and the progeny of these plants were segregated to test for complementation of the lethal phenotype of *toc75-III-1* homozygous plants. At least five independent lines were selected for each construct. The segregation analysis of representative T3 lines is shown in *Table 1*. As expected, *toc75-III-1*(+/-) progeny carrying the homozygous *TOC75* wild type transgene segregated at a near 3:1 ratio of hygromycin resistant:sensitive plants, consistent with complementation of the *toc75-III-1* insertion (*Baldwin et al., 2005*, *Shipman-Roston et al., 2010*) (*Table 1A*). Furthermore, plants homozygous for both *toc75-III-1* and *TOC75* alleles (*toc75-III-1*[+/+] *TOC75* [+/+]) could be recovered (*Table 1A*), indicating complementation of the lethal phenotype. By contrast, progeny of *toc75-III-1*(+/-) lines homozygous for the *TOC75ΔP1, TOC75ΔP1-2* or *TOC75ΔP1-3* deletion constructs segregated at 2:1 or lower ratios of hygromycin resistant:sensitive plants, indicating that the POTRA deletions were unable to complement *toc75-III-1* (*Table 1B*). No homozygous *toc75-III-1* lines expressing the POTRA-deletion constructs were recovered. On the basis of these data, we conclude that the POTRA domains are required for Toc75 function.

Heterozygous *toc75-III-1* plants are phenotypically indistinguishable from the wild-type plants (*Baldwin et al., 2005*) under normal growth conditions, implying that one copy of *TOC75-III* is sufficient for chloroplast function. Although most plants transformed with the Toc75 POTRA-deletion constructs (e.g. *TOC75ΔP1#1, TOC75ΔP1-2#1* or *TOC75ΔP1-3#1*; *Table 1B*) were phenotypically

**Table 1.** Segregation analysis of *toc75-III-1* plants transformed with full-length and POTRA deletion constructs of Toc75 expressed under the *TOC75* promoter.

| Parental plants | Hygromycin resistant (R) | Hygromycin sensitive (S) | R:S ratio | $\chi^2$- value[a] | $p_{0.05}$-value[b] |
|---|---|---|---|---|---|
| A. | | | | | |
| *toc75-III-1*(+/-) | 76 | 41 | 1.85 | 0.18 | 0.67 |
| *toc75-III-1*(+/-) *TOC75* (+/+) | 82 | 30 | 2.73 | 0.19 | 0.66 |
| *toc75-III-1*(+/+) *TOC75* (+/+) | 119 | - | - | | |
| B. | | | | | |
| *toc75-III-1*(+/-) *TOC75ΔP1*(+/+)#1 | 79 | 38 | 2.08 | 3.49 | 0.062 |
| *toc75-III-1*(+/-) *TOC75ΔP1-2*(+/+)#1 | 76 | 39 | 1.95 | 4.87 | 0.027 |
| *toc75-III-1*(+/-) *TOC75ΔP1-3*(+/+)#1 | 72 | 35 | 2.06 | 3.39 | 0.065 |
| C. | | | | | |
| *toc75-III-1*(+/-) *TOC75ΔP1*(+/+)#2 | 67 | 48 | 1.4 | 17.19 | 0.001 |
| *toc75-III-1*(+/-) *TOC75ΔP1-2*(+/+)#2 | 64 | 50 | 1.28 | 21.62 | <0.001 |
| *toc75-III-1*(+/-) *TOC75ΔP1-3*(+/-)#2 | 61 | 59 | 1.03 | 37.37 | <0.001 |
| D. | | | | | |
| *toc75-III-1*(+/-) *TOC75$_{myc}$*(+/+) | 112 | 38 | 2.95 | 0.009 | 0.92 |
| *toc75-III-1*(+/+) *TOC75$_{myc}$*(+/+) | 115 | - | - | | |
| *toc75-III-1*(+/-) *TOC75ΔP1$_{myc}$* (+/+) | 63 | 48 | 1.31 | 19.7 | <0.001 |

[a]Goodness-of-fit of the observed segregation ratios to the expected 2:1 ratio for *toc75-III-1* (+/-) or expected 3:1 ratio for complementation of *toc75-III-1* with the indicated *TOC75* gene constructs was assessed by $\chi^2$ analysis.

[b]p-Values were calculated using Graphpad Prism software version 4.00.

indistinguishable from *toc75-III-1* plants (data not shown), several transformed lines (*TOC75ΔP1#2, TOC75ΔP1-2#2* or *TOC75ΔP1-3#2*) showed progressively increased pale phenotypes and reduced growth rates (*Figure 1C*). The chlorophyll levels in *TOC75ΔP1#2 and TOC75ΔP1-2#2* plants were reduced by 26% and 62%, respectively (*Figure 1D*). *TOC75ΔP1-3#2* plants were nearly albino and exhibited severe growth defects (*Figure 1C*). Interestingly, the hygromycin resistance marker in progeny of the *TOC75ΔP1#2, TOC75ΔP1-2#2* and *TOC75ΔP1-3#2* lines segregated at a ratio significantly lower than the expected 2:1 ratio (*Table 1C*). These data suggest that expression of Toc75ΔP1, Toc75ΔP1-2 or Toc75ΔP1-3 proteins in these lines competed with the normal function of native Toc75, expressed from the single wild-type allele in the *toc75-III-1*(+/-background, resulting in disruption of the import apparatus and a consequent impact on plant growth and viability.

The differences in the segregation (*Table 1*) and phenotypes (*Figure 1C*) in the *TOC75ΔP1#2, TOC75ΔP1-2#2* and *TOC75ΔP1-3#2* plants compared to *TOC75ΔP1#1, TOC75ΔP1-2#1* and

**Table 2.** List of primers used for genotyping plants.

| Primers used for making POTRA deletions in *TOC75* genomic construct for generating transgenic plant | |
| --- | --- |
| For *TOC75ΔP1* | |
| Primer 1 | CTTAGTGGTTTCAAGAAGTATTGGCAATCTGCTGATAGG |
| Primer 2 | CCTATCAGCAGATTGCCAATACTTCTTGAAACCACTAAG |
| For *TOC75ΔP1-2* | |
| Primer 3 | CTTAGTGGTTTCAAGAAGTATATAACTCAGCTAGTTATTCAG |
| Primer 4 | CTGAATAACTAGCTGAGTTATATACTTCTTGAAACCACTAAG |
| For *TOC75ΔP1-3* | |
| Primer 5 | CTTAGTGGTTTCAAGAAGTATCAGAAGTCAGCTGAAGCT |
| Primer 6 | AGCTTCAGCTGACTTCTGATACTTCTTGAAACCACTAAG |
| Primers used for myc insertions in *TOC75* genomic construct for generating transgenic plant | |
| For *TOC75_myc* and *TOC75ΔP1_myc* using *TOC75* and *TOC75ΔP1* as templates, respectively | |
| Primer 7 | GATGAAGAACAAAAACTTATTTCTGAAGAAGATCTGGAACAATCACCGGAT |
| Primer 8 | ATCCGGTGATTGTTCCAGATCTTCTTCAGAAATAAGTTTTTGTTCTTCATC |
| Primers used for genotyping transgenic plants | |
| For *TOC75* | |
| Primer 9 | TTCTTTGATCGACGGAGAC |
| Primer 10 | CAGCAAACGAGATTGTAACACC |
| For *TOC75ΔP1* | |
| Primer 11 | GGTTTCAAGAAGTATTGGCAATCTGCTGAT |
| Primer 12 | GACATGTGTGTTCTTCACGGGTATTCTGATCTC |
| TOC75ΔP1-2 | |
| Primer 13 | GGTTTCAAGAAGTATATAACTCAGCTAGTT |
| Primer 14 | GACATGTGTGTTCTTCACGGGTATTCTGATCTC |
| For *TOC75ΔP1-3* | |
| Primer 15 | GGTTTCAAGAAGTATCAGAAGTCAGCTGAAGTC |
| Primer 16 | GACATGTGTGTTCTTCACGGGTATTCTGATCTC |
| Primers used for generating transit peptide deletions using *TOC75ΔP1* and *TOC75ΔP1-2* cDNA | |
| Primer 17 | GAAGGAGATATACATATGGATGAAGAACAATCACCGG |
| Primer 18 | CTACTTCTTGTTAGTGGCCCATATGTATATCTCCTTCTTAAAG |
| Primers used to construct POTRA domain constructs for solid phage binding assays | |
| Primer 19 | CTCGAGGGTGATGAAGAACAATCACCGG |
| Primer 20 | CTCGAGTTCTAGCTCCTTAAGCTTGATCTC |

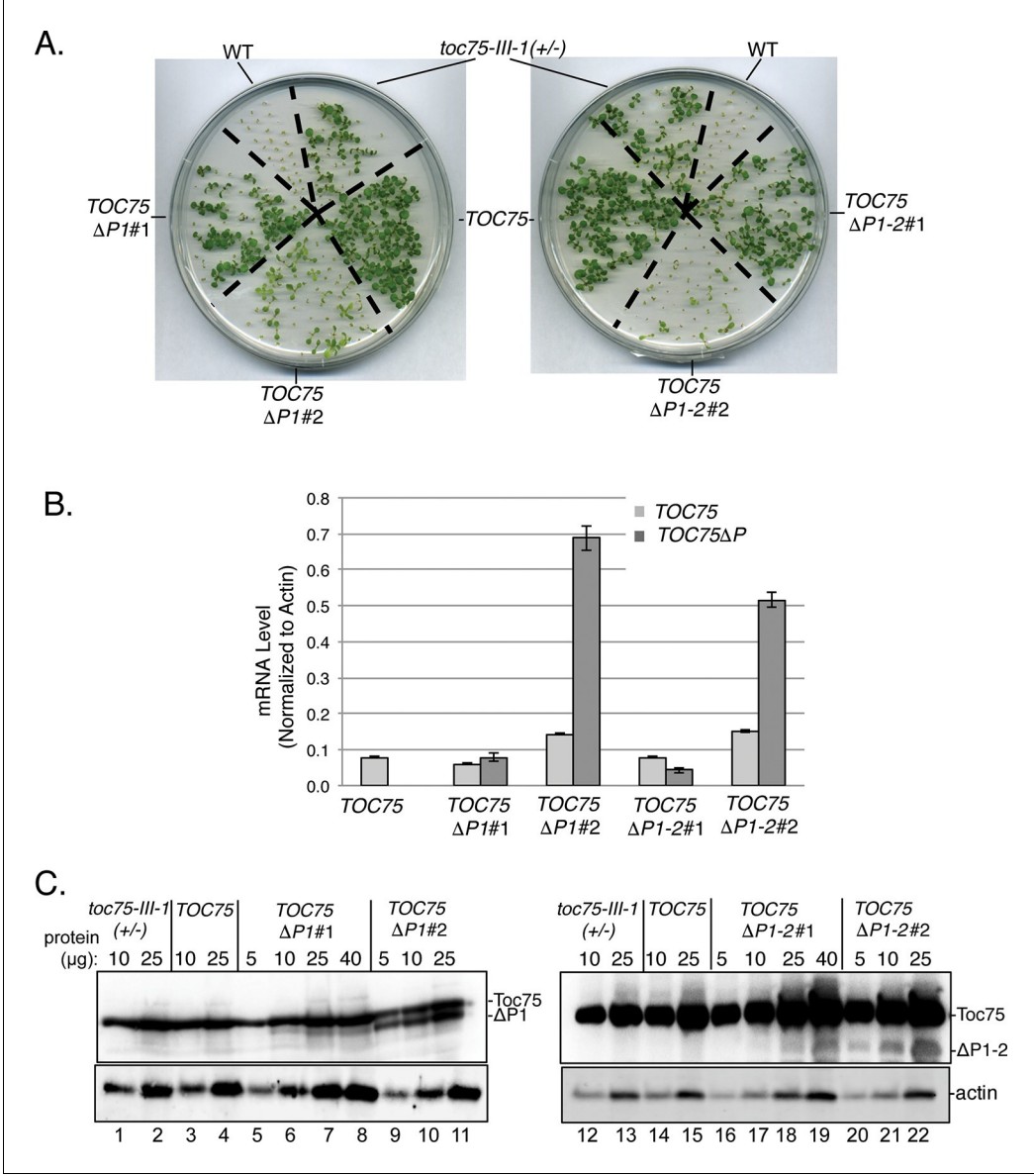

**Figure 2.** Dominant negative phenotypes exhibited by plants expressing Toc75ΔP1 or Toc75ΔP1-2. (**A**) Phenotypes of wild type, *toc75-III-1* (+/-) and *TOC75* plants compared to plants expressing lower (*TOC75ΔP1#1* and *TOC75ΔP1-2#1*) or higher (*TOC75ΔP1#2 and TOC75ΔP1-2#2*) levels of Toc75ΔP1 and Toc75ΔP1-2, respectively. Plants were grown for 14 days on agar media containing 20 μg/ml hygromycin. (**B**) Toc75, Toc75ΔP1 or Toc75ΔP1-2 mRNA levels in the transgenic plants indicated at the bottom of the graph as determined by quantitative real-time PCR. The primers used for RT-qPCR are listed in *Table 3*. Data represent the mean of three replicates. Error bars represent standard deviation. (**C**) Immunoblots of protein extracts from *TOC75, TOC75ΔP1#1, TOC75ΔP1-2#1, TOC75ΔP1#2 and TOC75ΔP1-2#2* plants using anti-atToc75 sera. The positions of Toc75, Toc75ΔP1 (ΔP1) or Toc75ΔP1-2 (ΔP1-2) are indicated to the right of the figure. Immunoblots of actin, a loading control, are shown in the bottom panel.

The following figure supplement is available for figure 2:

**Figure supplement 1.** Dominant negative phenotypes exhibited by plants expressing various levels of Toc75ΔP1 or Toc75ΔP1-2.

*TOC75ΔP1-3#1* plants, suggested that the levels of Toc75ΔP1, Toc75ΔP1-2 or Toc75ΔP1-3 in these lines varied. To further examine the nature of the aberrant segregation, we focused on *TOC75ΔP1#2, TOC75ΔP1-2#2* lines. We were unable to pursue further characterization of *TOC75ΔP1-3* because of the severe phenotype in lines expressing this construct (*Figure 1C*). We compared the expression levels of Toc75ΔP1 or Toc75ΔP1-2 in lines showing normal 2:1 hygromycin resistance segregation (*TOC75ΔP1#1 and TOC75ΔP1-2#1*) and those showing aberrant segregation (*TOC75ΔP1#2 and TOC75ΔP1-2#2*) (*Table 1B and C*). *TOC75ΔP1#1* and *TOC75ΔP1-2#1* plants were nearly indistinguishable from heterozygous *toc75-III-1* plants, whereas *TOC75ΔP1#2* and *TOC75ΔP1-2#2* plants showed a lower germination rate and significant growth defects when grown on agar plates (*Figure 2A*). Real-time reverse transcription quantitative PCR (RT-qPCR) (*Figure 2B*) and immunoblotting of extracts (*Figure 2C*) from seedlings of these lines indicated that the expression of the Toc75 POTRA deletion was higher in *TOC75ΔP1#2* and *TOC75ΔP1-2#2* plants compared to *TOC75ΔP1#1* and *TOC75ΔP1-2#1*. The examination of additional transformants confirmed that the severity of defects in growth (*Figure 1 – supplement 1*) and germination (*Figure 2 - supplement 1A and B*) correlated with the levels of Toc75ΔP1 or Toc75ΔP1-2 accumulation (*Figure 2 – supplement 1C*). Although we did not examine the reasons for the differences in expression between the lines, the location of transgene insertion in the Arabidopsis genome could account for the variation (*Gelvin and Kim, 2007*). These data indicate that increased expression of the POTRA deletions induces significant phenotypic defects, consistent with a dominant-negative effect on native Toc75 function in the *toc75-III-1*(+/-) background. Consistent with this observation, the levels of Toc75ΔP1 or Toc75ΔP1-2 mRNA in *TOC75ΔP1#2* or *TOC75ΔP1-2#2* plants were 3.5–4.5 times higher than those of native Toc75 mRNA, whereas the Toc75ΔP1 or Toc75ΔP1-2 protein levels were significantly lower than native Toc75. This suggests that expression of *TOC75ΔP1#2* and *TOC75ΔP1-2#2* is tightly

**Table 3.** List of primers used for RT-qPCR.

| TOC75 | |
|---|---|
| Primer 21 | TTCTTTGATCGACGGAGAC |
| Primer 22 | CAGCAAACGAGATTGTAACACC |
| TOC75ΔP1 | |
| Primer 23 | GGTTTCAAGAAGTATTGGCAATCTGCTGAT |
| Primer 24 | ACATCTGCATAACCTCACCATACA |
| TOC75ΔP1-2 | |
| Primer 25 | GGTTTCAAGAAGTATATAACTCAGCTAGTT |
| Primer 26 | GCGTGGATTGACTTCAATGTT |
| TOTAL TOC75 | |
| Primer 27 | AAGCTTGGTAATGTGGTTGAA |
| Primer 28 | TCAACAATAATGCCCCCTTC |
| TOC159 | |
| Primer 29 | AGAACCAACCAACCCCTTCT |
| Primer 30 | ACCAAATTCGGCTTCTCCTT |
| TOC33 | |
| Primer 31 | GGTGCAAAACCTTGCTTGTT |
| Primer 32 | GGAAGAGCCTTTTCGTCCTT |
| TIC22-III | |
| Primer 33 | AAAACATGAGTTATCGCCCTGT |
| Primer 34 | TTGCTCAGTTGAAACCTCAAAA |
| TIC22-IV | |
| Primer 35 | ATGCGTTAGAGCTCAAATCCTC |
| Primer 36 | CATCTCCATTTTCCTCAACACA |

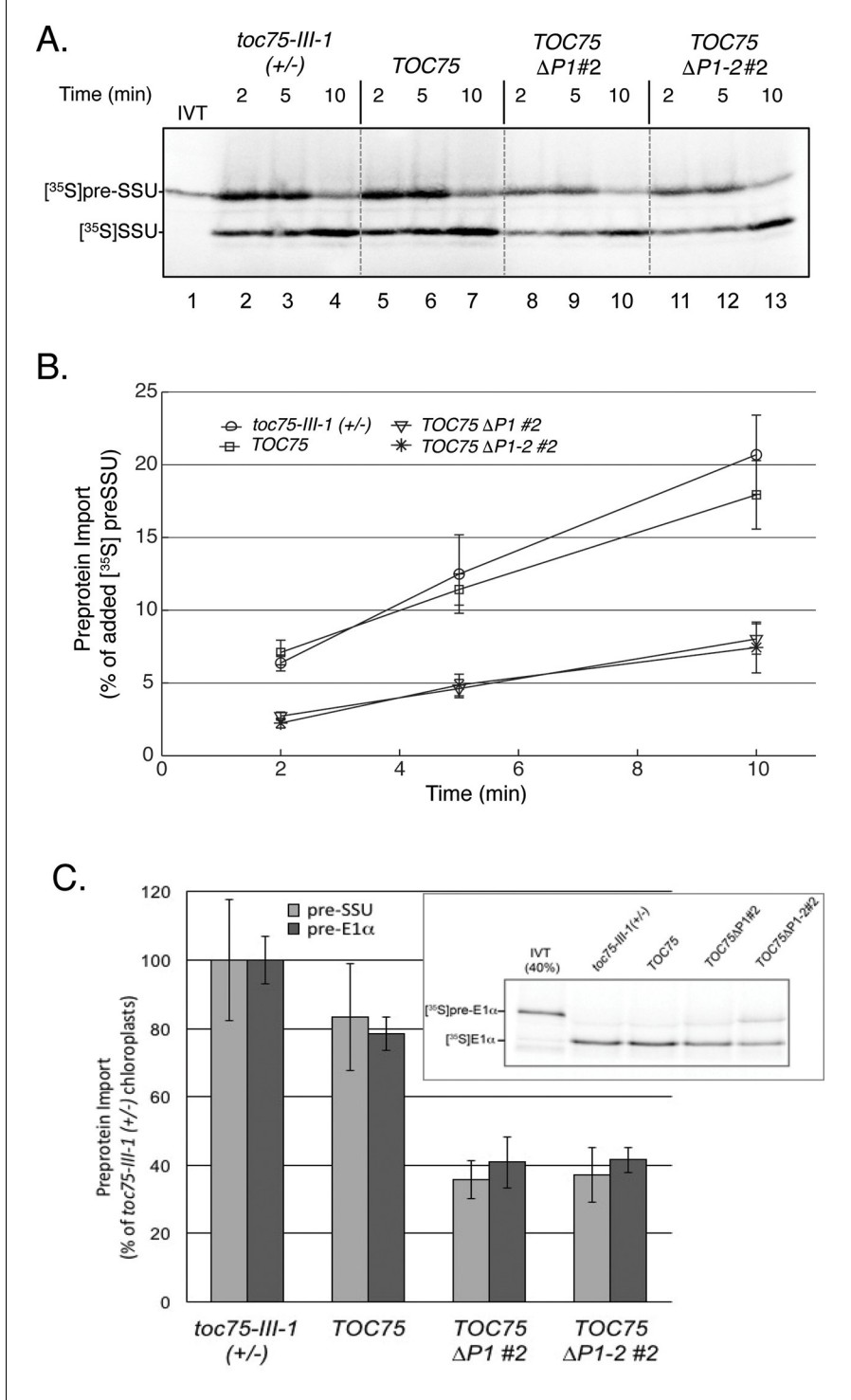

**Figure 3.** Preprotein import is decreased in chloroplasts isolated from *TOC75ΔP1#2 and TOC75ΔP1-2#2* plants compared to *toc75-III-1* (+/-) or *TOC75* plants. (**A**) In vitro import with isolated chloroplasts from *TOC75ΔP1#2 and TOC75ΔP1-2#2* plants. In vitro-translated [35S]preSSU was incubated with equivalent numbers of chloroplasts ($10^7$ chloroplasts) from *toc75-III-1* (+/-), *TOC75, TOC75ΔP1#2* and *TOC75ΔP1-2#2* plants under import conditions in the presence of 3 mM ATP for the times indicated. Lane 1 contains 10% of the [35S]preSSU added to each reaction (IVT). Dashed lines indicate that the panels in the figure were generated from different regions of the same SDS-PAGE gel using samples from the same experiment. The chloroplasts were analyzed directly by SDS-PAGE and phosphor imaging. (**B**) Quantitative analysis of the protein import assays in (**A**). Data represent the mean of

*Figure 3 continued on next page*

*Figure 3 continued*

triplicate experiments, with bars indicating standard error. (C) Comparison of the import of in vitro-translated [$^{35}$S] preSSUand [$^{35}$S]preE1α into isolated chloroplasts from *toc75-III-1 (+/-)*, *TOC75, TOC75ΔP1#2 and TOC75ΔP1-2#2* plants. Import is presented as a percentage of the import of each preprotein observed in *toc75-III-1 (+/-)* chloroplasts. [$^{35}$S]preSSU import data for the graph were derived from (A). The inset shows a representative gel of the triplicate experiments used to generate the graph for [$^{35}$S]preE1α. Lane 1 contains 40% of the [$^{35}$S]pre E1α added to each reaction (IVT).

controlled post-transcriptionally or by proteolytic degradation of Toc75ΔP1 or Toc75ΔP1-2, most likely in response to the negative impact of the POTRA deletions on chloroplast function.

## POTRA deletions disrupt protein import into chloroplasts

The TOC complex mediates the initial recognition of preproteins followed by their translocation across the outer envelope membrane via the Toc75 channel (*Richardson et al., 2014*). To test how deletion of the POTRA domains might impact preprotein translocation at the outer membrane, we measured time-dependent import kinetics in chloroplasts isolated from *toc75-III-1 (+/-)*, *TOC75*, *TOC75ΔP1#2 and TOC75ΔP1-2#2* plants. Isolated chloroplasts were incubated with [$^{35}$S]-labeled chloroplast preprotein, preSSU, in the presence of 3 mM ATP to promote import for 2, 5 or 10 min at 26° (*Figure 3A*). Quantitative analysis indicates that preprotein import rates in chloroplasts from *toc75-III-1 (+/-)* and *TOC75* plants were statistically indistinguishable (*Figure 3B*), consistent with the normal phenotype of these plants. By contrast, the levels of [$^{35}$S]preSSU import into chloroplasts expressing Toc75ΔP1, and Toc75ΔP1-2 were reduced to ~35% of the levels observed in chloroplasts from *toc75-III-1 (+/-)* and *TOC75* plants (*Figure 3B*). A similar reduction in import was observed for an additional preprotein, the precursor of the E1α subunit of pyruvate dehydrogenase ([$^{35}$S]preE1α) (*Figure 3C*). PreE1α utilizes an import pathway involving Toc75 and a set of TOC receptor GTPase isoforms distinct from those involved in preSSU import (*Ivanova et al., 2004*). These data demonstrate that Toc75ΔP1 and Toc75ΔP1-2 disrupt the function of protein import complexes involved in multiple import pathways, resulting in the defects in chloroplast biogenesis observed in *TOC75ΔP1#2 and TOC75ΔP1-2#2* plants.

To understand whether the reduced import in chloroplasts expressing Toc75ΔP1, and Toc75ΔP1-2 was due to defects in the recognition or translocation of preproteins at the outer membrane, we examined the interaction of [$^{35}$S]preSSU with the import machinery under two distinct energy conditions. In the absence of ATP, preproteins bind to chloroplasts in a low-affinity, energy-independent interaction involving the Toc159 and Toc33 GTPase receptors (*Aronsson and Jarvis, 2011*, *Inaba and Schnell, 2008*). In the presence of 0.1 mM ATP, preproteins are promoted to a higher affinity intermediate, which is inserted across the outer membrane and represents the initial translocation of preprotein across the envelope (*Fitzpatrick and Keegstra, 2001*, *Inaba and Schnell, 2008*). The distinction between energy-independent binding and the formation of the higher affinity intermediate is apparent by an increase in the levels of preprotein that stably associates with chloroplasts (*Olsen and Keegstra, 1992*). As expected *TOC75 and toc75-III-1(+/-)* chloroplasts showed 53% and 97% increases in chloroplast-associated preprotein, respectively, in the presence of 0.1 mM ATP compared to chloroplasts depleted of ATP (*Figure 4A and B*). By contrast, the stimulation of chloroplast-associated [$^{35}$S]preSSU in the presence of 0.1 mM ATP, in *TOC75ΔP1#2*, and *TOC75ΔP1-2#2* chloroplasts was ~25% and was statistically indistinguishable from energy-independent binding (*Figure 4A and B*). Incubation of the chloroplasts from the 0.1 mM ATP-binding experiment with 3 mM ATP to promote import of the intermediate into the stroma resulted in translocation and processing of 70–85% of bound [$^{35}$S]preSSU in the *TOC75 and toc75-III-1(+/-)* control chloroplasts and *TOC75ΔP1#2*, and *TOC75ΔP1-2#2* chloroplasts (*Figure 4A and C*). These data suggest that the POTRA deletions impact the transition from low-affinity, energy-independent binding to an energy-dependent, high-affinity insertion across the outer membrane, but not subsequent translocation across the inner envelope into the stroma.

The results from *Figure 4* suggest that the POTRA deletions do not have a major impact on initial energy-independent binding of preSSU, but significantly impact preprotein translocation across the outer membrane. To test this hypothesis, we investigated the kinetic parameters of import in

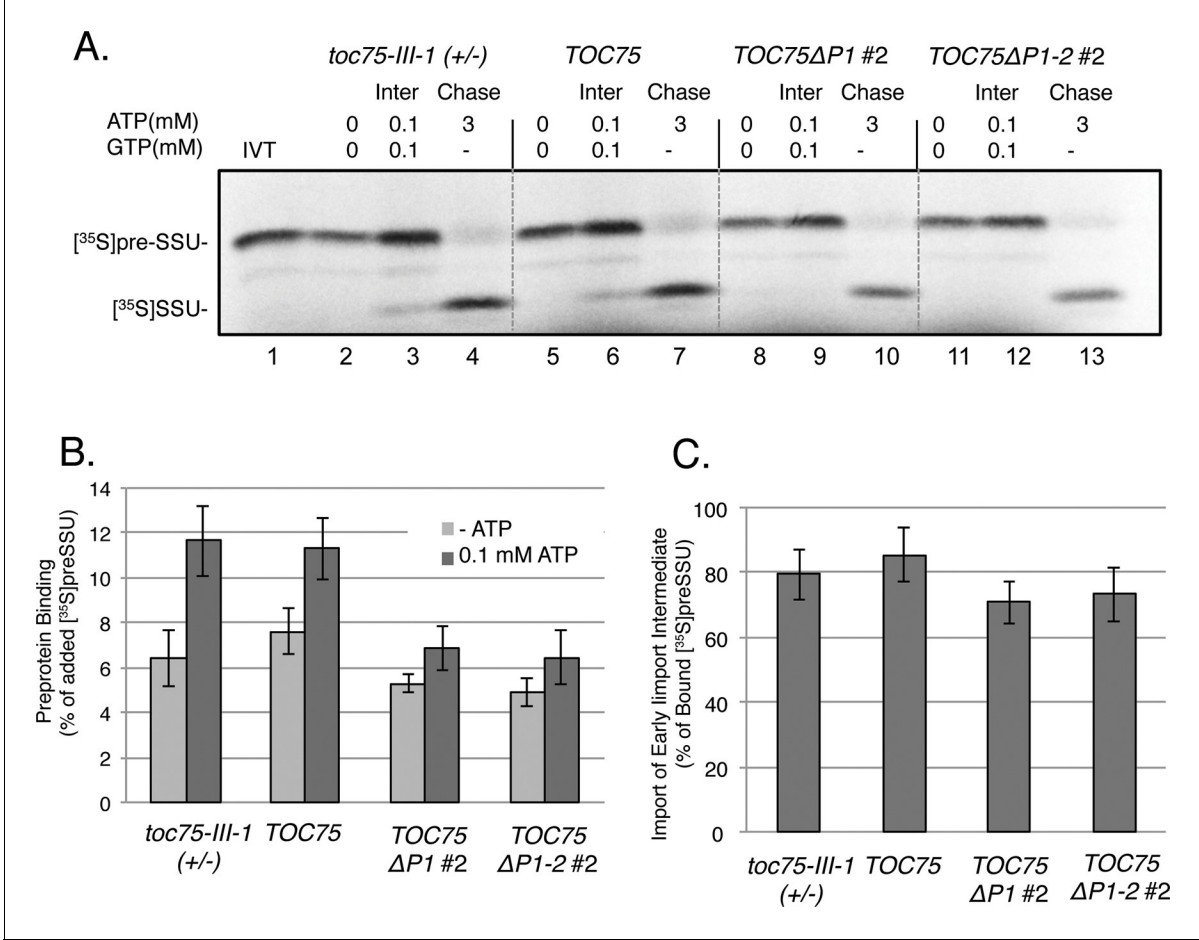

**Figure 4.** Deletion of POTRA-1 and -2 disrupt preprotein translocation across the outer membrane. (A) Energy-dependent and independent binding of [35S]preSSU to chloroplasts from *TOC75ΔP1#2 and TOC75ΔP1-2#2* plants. In vitro-translated [35S]preSSU was incubated at 26°C for 5 min with chloroplasts ($10^7$) from *toc75-III-1* (+/-), *TOC75, TOC75ΔP1#2 and TOC75ΔP1-2#2* plants in the absence of exogenous energy or in the presence of 0.1 mM ATP and GTP to promote translocation across the outer membrane and form an early import intermediate (Inter). *toc75-III-1* (+/-), *TOC75, TOC75ΔP1#2 and TOC75ΔP1-2#2* chloroplasts ($10^7$) containing bound early import intermediate (Inter) were reisolated and resuspended in the presence of 3 mM ATP to promote full translocation of the bound preprotein into the stroma (Chase). Lane 1 contains 10% of the in vitro translated [35S]preSSU added to each reaction (IVT). Dashed lines indicate that the figure was generated from different regions of the same SDS-PAGE gel using samples from the same experiment. The chloroplasts were analyzed directly by SDS-PAGE and phosphor imaging. (B) Quantitation of the [35S]preSSU early import intermediate from (A). (C) Quantitation of [35S]SSU imported from the chase experiment in (A). Data represent the mean of triplicate experiments, with bars indicating standard error.

*TOC75ΔP1#2* and *TOC75ΔP1-2#2* chloroplasts to determine the affinity and maximum translocation capacity of TOC translocons for preSSU. We generated a preprotein binding curve with increasing concentrations of purified *E. coli*-expressed preSSU-FLAG-HIS in the presence of 0.1 mM ATP and visualized bound preprotein by immunoblotting with anti-FLAG (*Figure 5A*). The apparent dissociation constants ($K_{d(app)}$) of preSSU-FLAG-HIS for TOC translocons in *TOC75* and *TOC75ΔP1#2* and *TOC75ΔP1-2#2* chloroplasts were 158.6 ± 29 nM, 187.7 ± 46 nM and 126.5 ± 24 mM, respectively (*Figure 5B*). This indicates that there were no measurable differences in the affinity of TOC translocons for preprotein in these plants. However, the maximum binding capacity was estimated as 91.4 ± 6.2, 57.1 ± 5.6 and 48.2 ± 3.3 fmol preSSU/µg chloroplast protein in *TOC75* and *TOC75ΔP1#2* and *TOC75ΔP1-2#2* chloroplasts, respectively (*Figure 5B*). As shown in *Figure 7*, the levels of TOC proteins were not reduced in the *TOC75ΔP1#2* and *TOC75ΔP1-2#2* plants, indicating that the reduced binding capacity was not due to a reduction in the levels of import components. These data are consistent with the hypothesis that deletion of the POTRA domains reduces the number of active TOC

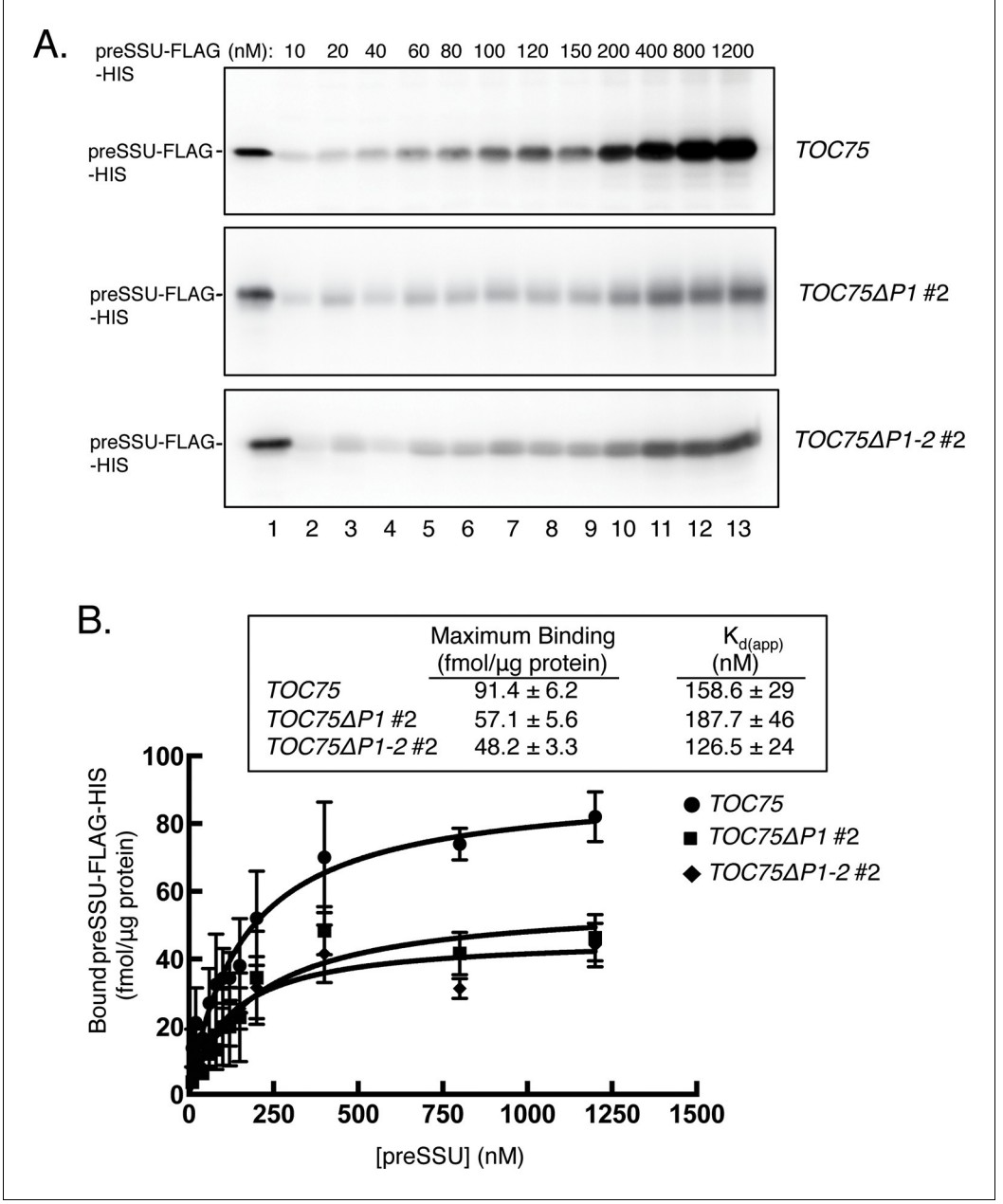

**Figure 5.** Expression ofToc75ΔP1 and Toc75ΔP1-2reduces the number of import-competent TOC complexes. (**A**) Saturation binding of preprotein in chloroplasts from *TOC75ΔP1#2 and TOC75ΔP1-2#2* plants. Isolated chloroplasts ($10^7$) from *TOC75, TOC75ΔP1#2 and TOC75ΔP1-2#2* plants were incubated with increasing amounts of *E. coli*-expressed, urea-denatured preSSU-FLAG-HIS in the presence of 0.1 mM ATP and GTP at 26°C for 5 min, to promote formation of an early import intermediate. Chloroplasts were reisolated through Percoll silica gel, resolved by SDS-PAGE, and the early import intermediate form of preSSU-FLAG-HIS was detected by immunoblotting with anti-FLAG. Lane 1 contains 0.64 pmol of the preSSU-FLAG-HIS. (**B**) Quantitation of the data from (**A**). Saturation binding analysis of the data in (**A**) is presented in the table inset. The maximum number of binding sites (Maximum Binding) and apparent $K_d$ were calculated by nonlinear fitting of the data in (**A**). Each data bar represents the mean ± SD (n = 3).

complexes at the outer envelope by interfering with their ability to mediate outer membrane translocation.

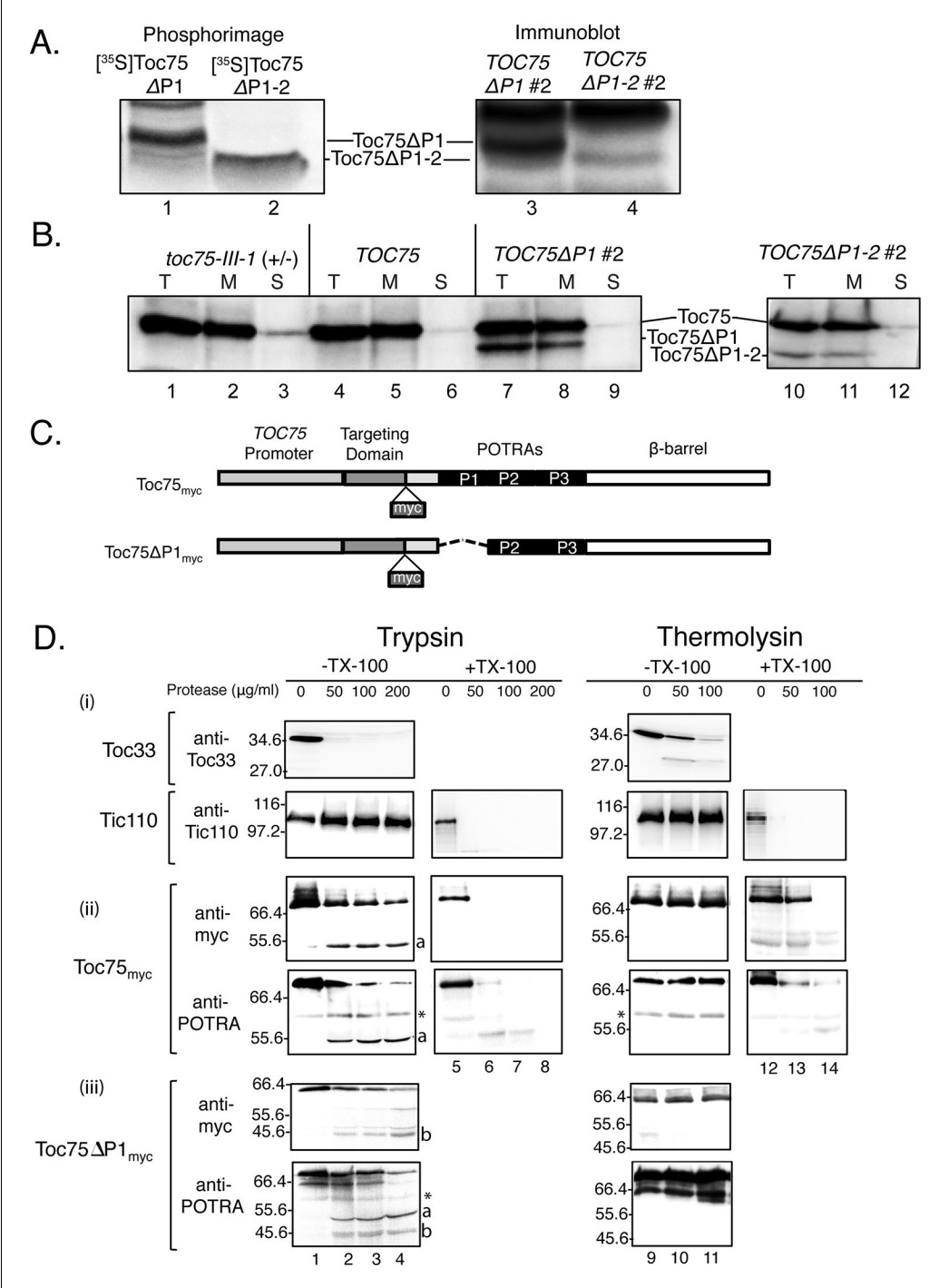

**Figure 6.** Toc75ΔP1 and Toc75ΔP1-2are properly targeted to the outer membrane with the POTRA domains oriented to the intermembrane space. (**A**) Toc75ΔP1 and Toc75ΔP1-2 accumulate as their mature forms in *TOC75ΔP1#2 and TOC75ΔP1-2#2* plants. In vitro translated mature [35S]Toc75ΔP1 and [35S]Toc75ΔP1-2 were mixed with chloroplasts extracts from *TOC75ΔP1#2 and TOC75ΔP1-2#2* plants and analyzed by SDS-PAGE. The mobility of [35S] Toc75ΔP1 and [35S]Toc75ΔP1-2, detected by phosphorimaging (Lanes 1 and 2), was compared to the mobility of endogenous Toc75ΔP1 and Toc75ΔP1-2, detected by immunoblotting with anti-atToc75 serum (Lanes 3 and 4). (**B**) Toc75ΔP1 and Toc75ΔP1-2 are integrated into the chloroplast envelope. Isolated chloroplasts (T) from heterozygous *toc75-III-1*, *TOC75, TOC75ΔP1#2 and TOC75ΔP1-2#2* plants were hypotonically lysed and fractionated by centrifugation at 18,000 × g for 30 min at 4°C into membrane pellet (M) and soluble (S) fractions. Equivalent samples of each fraction from chloroplasts corresponding to 10 µg chlorophyll was resolved by SDS-PAGE and immunoblotted with anti-atToc75 serum. (**C**) Toc75myc and Toc75ΔP1myc constructs used in this study. The dashed line represents the deleted POTRA domain region in POTRA1-deleted constructs. The site of insertion of myc tag is shown in each panel. (**D**) Protease sensitivity of Toc75myc, and Toc75ΔP1myc proteins in isolated intact chloroplasts. Intact chloroplasts from *TOC75,*

Figure 6 continued

TOC75$_{myc}$ and TOC75ΔP1$_{myc}$ seedlings were treated with trypsin or thermolysin in the absence (-) or presence (+) of 1% Triton X-100. Reactions were incubated on ice for 30 min, and proteolysis was stopped with 2.5 mM PMSF, 0.05 mg/mL Nα-Tosyl-L-lysine chloromethyl ketone (TLCK), 0.25 mg/mL soybean trypsin inhibitor, and 2 µg/mL aprotinin (for trypsin) or 20 mM EDTA (for thermolysin). Chloroplasts were analyzed by immunoblotting with antibodies against various proteins as indicated. The asterisk indicates the position of a non-specific immunoreactive band. a and b denote bands corresponding to the ~55 kDa and ~46 kDa trypsin fragments of Toc75$_{myc}$ and Toc75ΔP1$_{myc}$, respectively.

## Effects of POTRA deletions on Toc75 targeting and membrane topology

To investigate the nature of the import defect in *TOC75ΔP1#2* and *TOC75ΔP1-2#2* plants, we examined whether the POTRA deletions disrupted Toc75 targeting and membrane integration. Toc75 is unique among known chloroplast outer envelope membrane proteins in being targeted to the membrane via a cleavable N-terminal bipartite targeting signal (*Tranel and Keegstra, 1996*). Toc75 is translated as an 89.2 kDa precursor (pre-Toc75) and processed sequentially at the outer membrane and intermembrane space, to generate an 84.6 kDa intermediate form and the 75 kDa mature form, respectively (*Inoue et al., 2001*, *Inoue and Keegstra, 2003*, *Shipman and Inoue, 2009*). Upon complete processing, the predicted sizes of mature Toc75ΔP1 and Toc75ΔP1-2 are 66.6 kDa and 52.7 kDa, respectively. To determine if the POTRA deletions were processed normally, we synthesized [$^{35}$S]Toc75ΔP1 and [$^{35}$S]Toc75ΔP1-2 in a cell-free in vitro translation system, mixed them with protein extracts from *TOC75ΔP1#2* and *TOC75ΔP1-2#2* plants and compared the sizes of the radiolabeled and endogenous Toc75ΔP1 and Toc75ΔP1-2 by phosphorimaging and immunoblotting, respectively. The results in *Figure 6A* show that the SDS-PAGE migration of the radiolabeled and endogenous Toc75ΔP1 and Toc75ΔP1-2 are coincident, indicating that the POTRA-deletions were processed to their mature forms. Consistent with proper processing, fractionation of chloroplasts into soluble and membrane fractions demonstrated that Toc75ΔP1 and Toc75ΔP1-2 from *TOC75ΔP1#2* and *TOC75ΔP1-2#2* plants, respectively, were membrane integrated (*Figure 6B*).

To test if the POTRA deletions impacted the topology of Toc75 in the outer membrane, we examined the sensitivity of full-length Toc75 and Toc75ΔP1 to exogenous protease in isolated, intact chloroplasts. We were unable to examine Toc75ΔP1-2 due to relatively low levels of expression of this construct in transgenic plants and the consequent inability to reliably detect proteolytic fragments from the deletion construct. To facilitate our studies, we expressed Toc75 and Toc75ΔP1 containing a myc epitope inserted just upstream of the POTRA domains at the N-terminus of the mature proteins in *toc75-III-1* plants (*Figure 6C*). The *TOC75$_{myc}$* transgene was able to fully complement the *toc75-III-1* lethal phenotype (*Table 1D*). Consistent with the *TOC75ΔP1* transformants, the *TOC75ΔP1$_{myc}$* lines showed both normal and pale phenotypes, and none were able to complement *toc75-III-1* (*Table 1D*). For subsequent studies, we selected a line, which showed a pale phenotype *TOC75ΔP1$_{myc}$* (data not shown) similar to *TOC75ΔP1#2*.

Isolated chloroplasts from *TOC75$_{myc}$* and *TOC75ΔP1$_{myc}$* plants were treated with various concentrations of trypsin or thermolysin, resolved by SDS-PAGE and immunoblotted with anti-myc or antibodies specific to the Toc75 POTRA domains. Previous studies have shown that Toc75 is resistant to low-to-moderate levels of thermolysin, whereas trypsin generates several proteolytic fragments, including peptides corresponding to the N-terminal POTRA deletions (*Tranel and Keegstra, 1996*, *Inoue and Potter, 2004*, *Sveshnikova et al., 2000*). As controls for protease activity, we examined the sensitivity of Toc33 and Tic110 to the protease treatments. Toc33 is anchored in the outer membrane, with a short C-terminal transmembrane helix and the bulk of the protein is exposed to the cytoplasm (*Kessler et al., 1994*), and Tic110 is anchored to the inner membrane, with the bulk of the protein extending into the stroma (*Li and Schnell, 2006*). As predicted, Toc33 is digested by both trypsin and thermolysin, with complete degradation of the cytoplasmic domain observed at 200 µg/ml trypsin and 100 µg/ml thermolysin (*Figure 6D* i). By contrast, Tic110 is completely resistant to the same concentrations of trypsin or thermolysin unless the membrane is disrupted with triton X-100 (*Figure 6D* i).

Toc75$_{myc}$ is degraded by treatments with increasing concentrations of trypsin (*Figure 6D* ii). Reduction in full length Toc75$_{myc}$ (75 kDa) is concomitant with the appearance of a ~55 kDa protease-resistant fragment, which reacts with anti-POTRA and anti-myc (*Figure 6D* ii, band a). Disruption

of the membrane with triton X-100 results in complete degradation of Toc75, indicating that the protected fragment observed in intact chloroplasts is not intrinsically resistant to proteolysis (*Figure 6D* ii). Treatment of chloroplasts from *TOC75ΔP1*$_{myc}$ plants with trypsin resulted in the degradation of both the 75 kDa Toc75 and the 68 kDa Toc75ΔP1$_{myc}$ proteins (*Figure 6D* iii), and generated two anti-POTRA reactive proteolytic fragments of ~55 kDa and ~46 kDa (*Figure 6D* iii, bands a and b). The difference in the size of the two anti-POTRA reactive fragments corresponds to the difference in size between native Toc75 and Toc75ΔP1 (8.6 KDa). The ~46 kDa fragment reacted with anti-myc (*Figure 6D* iii, band b), confirming its identity as an N-terminal fragment containing the Toc75ΔP1$_{myc}$ POTRA domain fragment. These data demonstrate that the peptides are generated from the N-terminal region of Toc75 and Toc75ΔP1$_{myc}$, respectively, and confirm that Toc75ΔP1 has the same topology as native Toc75 in the outer envelope membrane.

## Protein import components accumulate to higher levels in plants expressing POTRA deletions

Toc75 is implicated in the insertion of Toc33 and Toc159 into the outer membrane (*Richardson et al., 2014*), and it participates in the initial stages of targeting of TIC components to the inner membrane (*Shi and Theg, 2013*). Consequently, we examined the expression and accumulation of TOC and TIC components in *TOC75ΔP1#2 and TOC75ΔP1-2#2* plants to determine if the import defects could result from a disruption in the levels of import components. Immunoblots indicated that overall levels of Toc75, including Toc75, Toc75ΔP1 and Toc75ΔP1-2, were increased up to two fold in plants expressing POTRA-deleted Toc75 compared to wild type or *TOC75* plants (*Figure 7A and B*). Levels of Toc33 protein were significantly increased (~five fold) in *TOC75ΔP1* and *TOC75ΔP1-2* plants, suggesting that accumulation of Toc33 is increased in response to expression of the POTRA deletion constructs (*Figure 7A and B*). Toc159 levels were slightly increased (<1.4-fold), but the increase was not statistically significant compared to *toc75-III-1* (+/-) plants (*Figure 7A and B*). The levels of OEP80, another member of Omp85 superfamily in the outer membrane of chloroplasts (*Hsu et al., 2012*, *Day et al., 2014*), were unchanged in plants expressing Toc75ΔP1 and Toc75ΔP1-2 (*Figure 7A and B*), further indicating that the increased accumulation of Toc33 was not due to a general increase in outer membrane protein levels. A representative TIC component, Tic110, did not show a change in abundance in *TOC75ΔP1#2* and *TOC75ΔP1-2#2* plants (*Figure 7A and B*). However, the levels of Tic22-III and Tic22-IV, two homologous protein import components of the intermembrane space (*Kouranov et al., 1998*, *Kasmati et al., 2013*), were increased in *TOC75ΔP1* and *TOC75ΔP1-2* plants. Tic22-IV levels were moderately but significantly increased, whereas Tic22-III showed a ~three fold increase in both deletion lines (*Figure 7A and B*).

We performed RT-qPCR to determine if the increases in protein accumulation observed in the immunoblotting experiments correlated with increased transcription. Compared to *toc75III-1* (+/-) and *TOC75* plants, the levels of total transcripts encoding Toc75 and the POTRA deletions (Toc75, Toc75ΔP1 and Toc75 ΔP2) were increased by four- and six fold in *TOC75ΔP1#2* and *TOC75ΔP1-2#2* plants, respectively (*Figure 7C*). Toc33 transcript also increased, as did those of Tic22-III and Tic22-IV (*Figure 7C*), whereas Toc159 transcript levels were unchanged (*Figure 7C*). These data suggest that the changes observed in the levels of import components in *TOC75ΔP1#2* and *TOC75ΔP1-2#2* plants were at least partially due to changes in gene expression in response to expression of the Toc75 POTRA deletions.

## POTRA deletions disrupt TOC complex stoichiometry

Toc75 forms stable complexes with Toc33 and Toc159 in stoichiometric ratios estimated at 4:4:1 or 3:3:1 (Toc75:Toc34:Toc159), based on the mobility of TOC complexes on blue-native gel electrophoresis (*Schleiff et al., 2003*, *Kikuchi et al., 2006*, *Chen and Li, 2007*). The over-accumulation of Toc75 and Toc33 in the POTRA deletion lines (*Figure 7*) suggested that the mutations might disrupt the stoichiometry of TOC complexes, thereby resulting in the observed import defects (*Figure 3*). As a first step to determine if Toc75ΔP1 and Toc75ΔP1-2 were interacting with other TOC components, we immunoprecipitated detergent-solubilized chloroplast membranes from *TOC75, TOC75ΔP1#2* and *TOC75ΔP1-2#2* plants with anti-Toc33-Sepharose and immunoblotted the samples with TOC antibodies. Toc159, full-length Toc75 and Tic110, a component of TIC complexes, co-immunoprecipitated with Toc33 from *TOC75, TOC75ΔP1#2* and *TOC75ΔP1-2#2* chloroplasts.

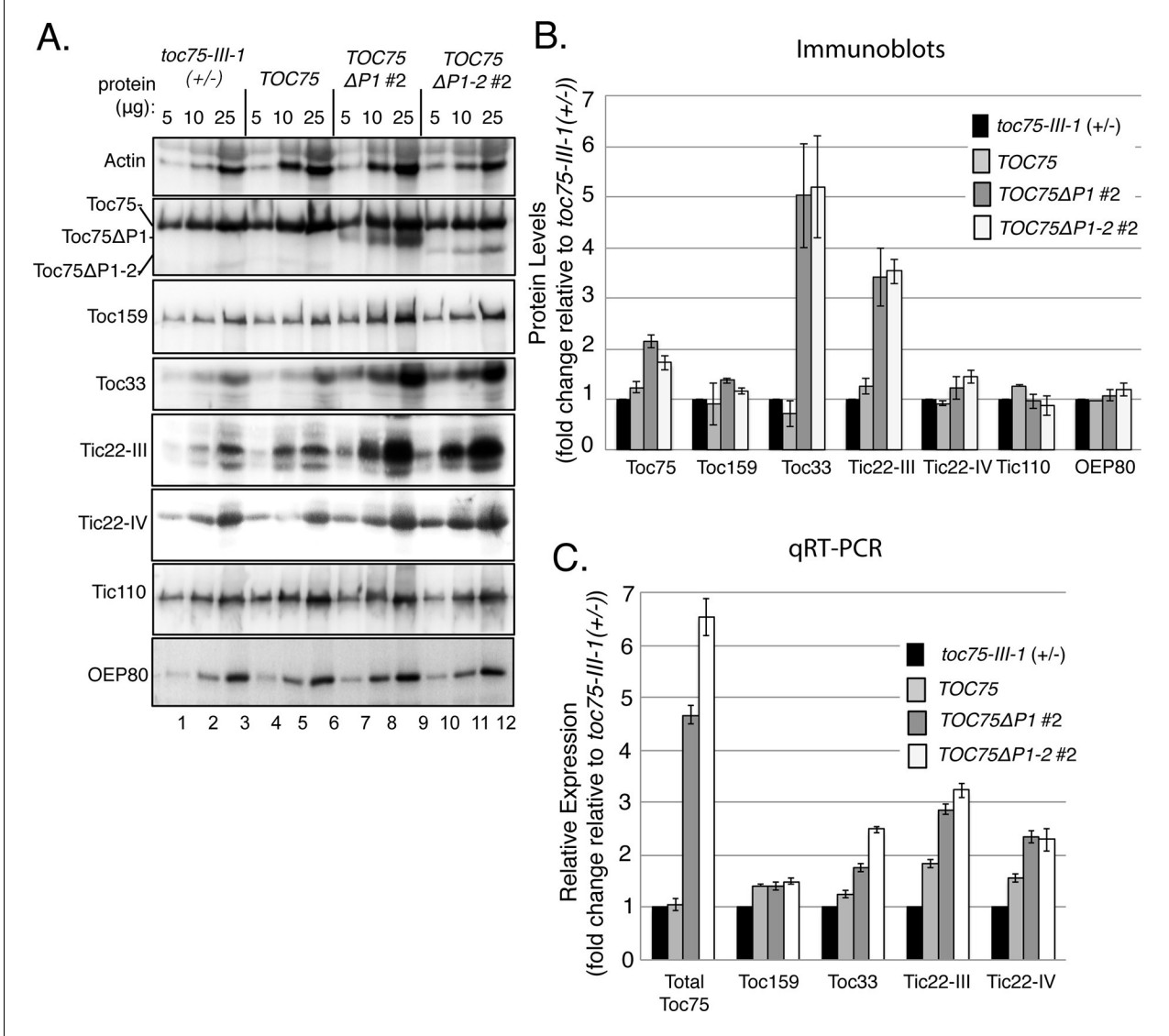

**Figure 7.** Accumulation of outer envelope proteins in *TOC75ΔP1#2 and TOC75ΔP1-2#2* plants. (**A**) Serial dilutions of protein extracts from *toc75-III-1* (+/-), *TOC75, TOC75ΔP1#2 and TOC75ΔP1-2#2* plants were immunoblotted using antisera corresponding to the proteins indicated at the left. (**B**) Relative quantitation of plastid proteins. The signal intensities falling within the linear range of chemiluminescence detection were normalized to the signal for actin in each sample and plotted as fold change relative to the levels of the corresponding protein in *toc75-III-1* (+/-) plants. (**C**) Relative expression of various proteins analyzed by RT-qPCR from *toc75-III-1* (+/-), *TOC75, TOC75ΔP1#2 and TOC75ΔP1-2#2* plants. The values are normalized to the internal levels of actin mRNA and plotted as the fold change relative to the levels of the corresponding mRNA in *toc75-III-1* (+/-) plants. Each error bar represents the mean ± SD (n = 3).

Toc75ΔP1 and Toc75ΔP1-2 also co-immunoprecipitate with Toc33 (*Figure 8A*). OEP80, an outer membrane protein that is not associated with TOC complexes, was not co-immunoprecipitated with anti-Toc33 (*Figure 8A*). These data indicate that Toc75ΔP1 and Toc75ΔP1-2 can interact directly or indirectly with other import components of TOC-TIC complexes.

To determine if the interactions detected in *Figure 8A* correspond to the correct stoichiometric assembly of TOC components, we analyzed the complexes by two-dimensional (2D) blue-native PAGE. As shown in *Figure 8B* i, Toc33, Toc75 and Toc159 from *TOC75* chloroplasts co-migrated in complexes, which peaked at ~1.3 MDa (region a). A second complex, with a peak at ~440 kDa (region b) migrated between 400 and 880 kDa and contained Toc75 and Toc33, but lacked detectable levels of Toc159. The presence of the ~1.3 MDa and ~440 kDa complexes is consistent with the

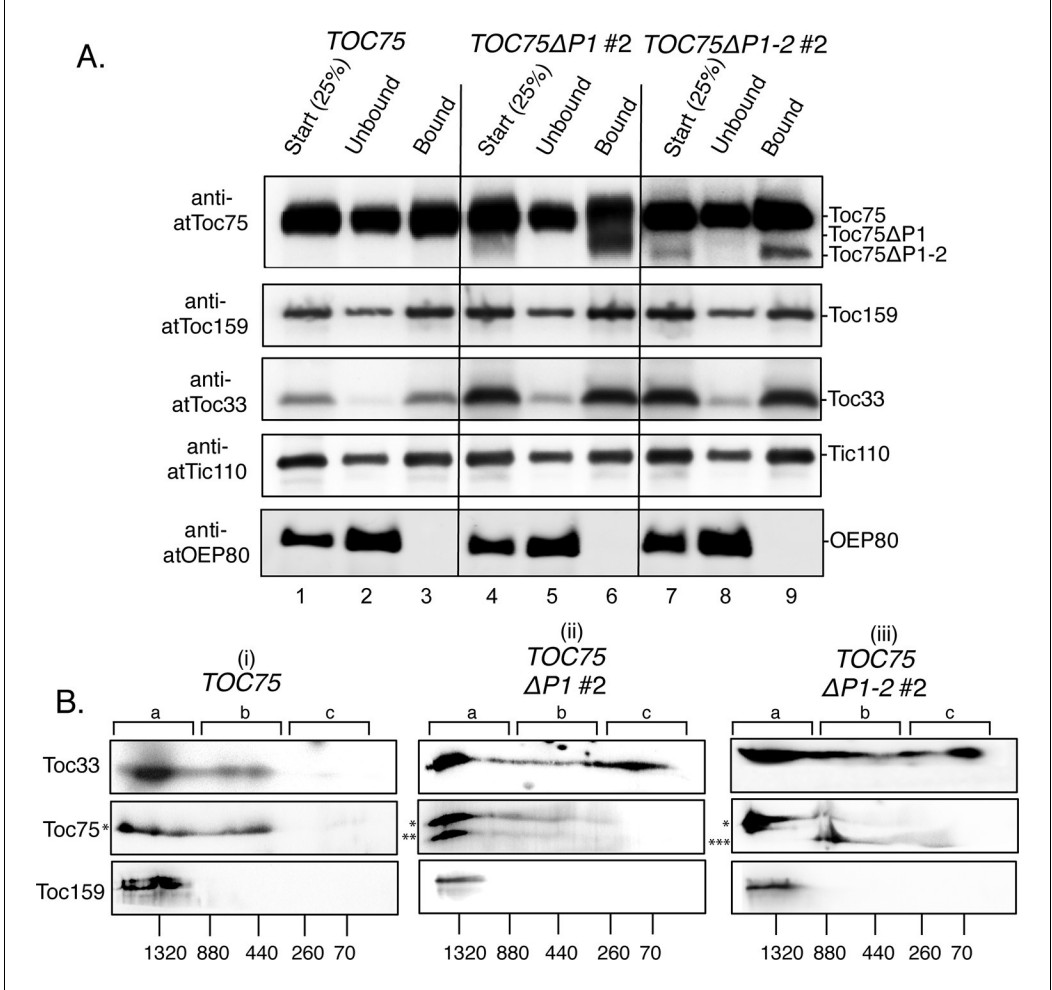

**Figure 8.** Toc75ΔP1 and Toc75ΔP1-2 interfere with the stoichiometric assembly of TOC complexes. (**A**) Co-immunoprecipitation of Toc75ΔP1 and Toc75ΔP1-2 with Toc33 and Toc159. Detergent-soluble chloroplast membranes from *TOC75, TOC75ΔP1#2 and TOC75ΔP1-2#2* plants were subjected to immunoaffinity chromatography on anti-atToc33 Sepharose. Twenty-five percent of the total extracts (Start) and unbound fractions (Unbound) or the eluate fractions (Bound) were resolved by SDS-PAGE and immunoblotted with antisera indicated to the left of each panel. (**B**) 2D Blue-native PAGE analysis of chloroplast membranes from *TOC75, TOC75ΔP1#2 and TOC75ΔP1-2#2* plants. Chloroplast membranes from *TOC75* (i), *TOC75ΔP1#2* (ii) *and TOC75ΔP1-2#2* (iii) plants were subjected to 4–12% BN-PAGE in the first dimension followed by 5–12% SDS-PAGE in the second dimension. Proteins were then transferred to a PVDF membrane and immunoblotted with the antisera indicated to the left of each panel. The positions of the major 1.3 MDa (region a), 440 kDa (region b) and 70 kDa (region c) complexes are indicated at the top of each panel. Asterisks to the left of the middle panels indicate the positions of the bands corresponding Toc75 (*), Toc75ΔP1 (**) and Toc75ΔP1-2 (***).

800–1000 kDa and 300–450 kDa TOC complexes previously reported from pea chloroplasts (*Kikuchi et al., 2006*) on 2D blue-native gels. The minor differences in the mobilities of the TOC complexes reported here and in the previous study are likely due to differences in the sizes of TOC components in Arabidopsis versus pea chloroplasts. Both 1.3 MDa and 440 kDa complexes containing native Toc75, Toc33 and Toc159 were also observed in *TOC75ΔP1#2* and *TOC75ΔP1-2#2* plants (*Figure 8B*). Toc75ΔP1 is detected in both 1.3 MDa and 440 kDa complexes in chloroplasts from *TOC75ΔP1#2* plants (*Figure 8B* ii), suggesting that Toc75ΔP1 assembles with the core TOC GTPase receptors. By contrast, Toc75ΔP1-2 was absent from the 1.3 MDa complexes and present only in complexes with a mobility similar or smaller than the 440 kDa complexes in *TOC75ΔP1-2#2* plants (*Figure 8B* iii). In conjunction with the observation that Toc75ΔP1-2 co-immunoprecipitates with Toc33 (*Figure 8A*), the results from 2D blue-native gels suggest that Toc75ΔP1-2 is able to interact with Toc33 but is unable to assemble into the 1.3 MDa complexes containing all three core TOC components.

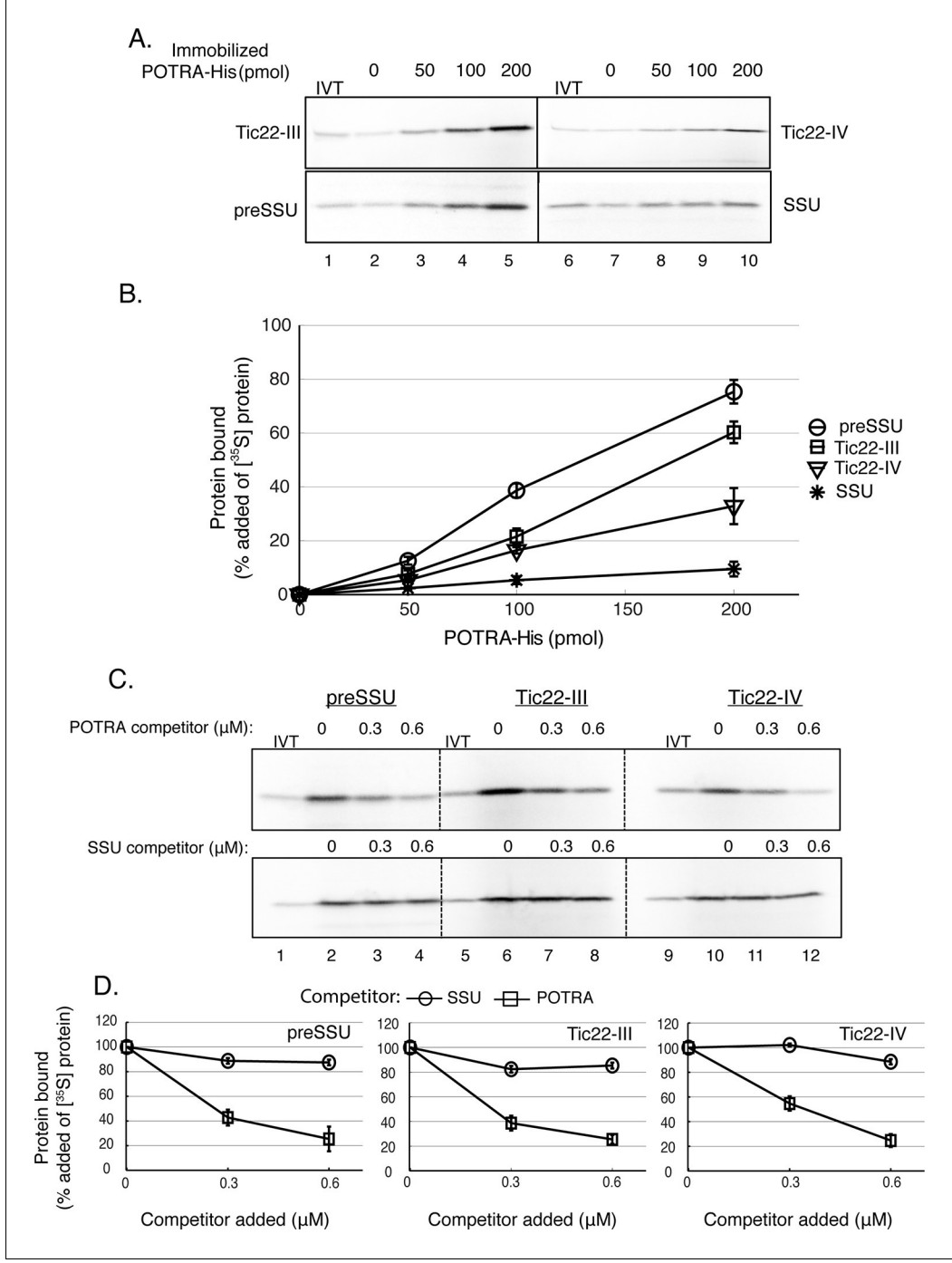

**Figure 9.** Toc75 POTRA domains interact directly with chloroplast preproteins and Tic22 isoforms. (**A**) Binding of Tic22-III, Tic22-IV and preSSU to immobilized Toc75 POTRA domains. [35S]-labeled Tic22-III, Tic22-IV, preSSU and SSU were incubated with increasing amounts of Ni-NTA resin-immobilized Toc75 POTRA-His. Lane 2 in A shows the background binding of radiolabeled proteins to the Ni-NTA resin in the absence of POTRAs. In all cases, binding to Ni-NTA resin alone (lane 2) was less than 10% of added radiolabelled protein. (**B**) Quantitation of the binding in (**A**). Binding is presented as the percentage of added [35S]-labeled proteins recovered in each reaction after subtracting binding to Ni-NTA resin alone (lane 2). (**C**) Binding of POTRA domains to preSSU, Tic22-III or Tic22-IV is specific. Two hundred picomole immobilized POTRA-His was incubated with [35S]-labeled preSSU, Tic22-III or Tic22-IV in the absence or presence of increasing concentrations of soluble POTRA domains or SSU as a competitor. (**D**) Quantitation of the binding in (**C**). Binding is presented as the percentage of maximal initial

*Figure 9 continued on next page*

*Figure 9 continued*

binding. Each data bar represents the mean ± SEM (n = 3). Lanes labeled IVT in panels A and C contain 10% of the in vitro translation product added to each reaction.

Interestingly, the levels of Toc33 migrating at a low molecular weight (~70 kDa peak) were increased significantly in *TOC75ΔP1#2* and *TOC75ΔP1-2#2* chloroplasts compared to *TOC75* chloroplasts (*Figure 8B*, region c). Neither Toc75, the Toc75 POTRA deletions or Toc159 co-migrate with this form of Toc33, suggesting that a considerable proportion of Toc33 is not assembled into native TOC complexes in the *TOC75ΔP1#2* and *TOC75ΔP1-2#2* chloroplasts. The dramatic increase in Toc33 expression relative to the other TOC components in *TOC75ΔP1#2* and *TOC75ΔP1-2#2* chloroplasts (*Figure 7*) is likely to be responsible, at least in part, for the Toc33 that is not assembled into TOC complexes (*Figure 8B*). Taken together, these data demonstrate that deletion of both the first and second POTRA domains of Toc75 disrupts assembly of native TOC complexes within the outer membrane.

## Toc75 POTRA domains interact with an intermembrane import component, Tic22

The increase in accumulation of the Tic22 isoforms in *TOC75ΔP1#2* and *TOC75ΔP1-2#2* plants, raised questions regarding the relationship of these two import proteins. Previous work had demonstrated a potential interaction between the N-terminal POTRA domains of an Omp85 family member in the outer membrane of the cyanobacteria, *Anabeana sp. PCC 7120*, and a chaperone protein in the periplasmic space, which is structurally related to Tic22 (*Tripp et al., 2012*). These observations led us to examine a possible physical interaction between Toc75 and Tic22 in the chloroplast intermembrane space. A protein fragment encompassing all three Toc75 POTRA domains and a C-terminal hexahistidine tag (POTRA-His) was expressed and purified from *E. coli* and tested for binding to Tic22-III and Tic22-IV. The recombinant POTRA-His was immobilized on nickel-nitrilotriacetic acid (Ni-NTA) matrix and incubated with in vitro–translated [$^{35}$S]Tic22-III and [$^{35}$S]Tic22-IV. Binding was measured as the fraction of [$^{35}$S]Tic22-III and [$^{35}$S]Tic22-IV that co-sedimented with the immobilized POTRA domains. Transit peptide-dependent binding of preSSU to Toc75 POTRAs has been demonstrated previously (*Ertel et al., 2005*), and therefore, we used [$^{35}$S]preSSU and [$^{35}$S]SSU as positive and negative controls for binding, respectively.

As shown in *Figure 9*, [$^{35}$S]Tic22-III, [$^{35}$S]Tic22-IV and [$^{35}$S]preSSU bound to immobilized POTRA-His in a dose-dependent manner, with maximum binding observed at 60%, 35% and 77% of added radiolabeled proteins, respectively (*Figure 9A and B*). None of the proteins exhibited significant binding to the Ni-NTA matrix alone (*Figure 9 A*, lanes 2 and 7), and <10% of added [$^{35}$S]SSU, the negative control, associated with the resin at the highest concentrations tested (*Figure 9B*). Soluble POTRA domains effectively competed for binding of [$^{35}$S]Tic22-III, [$^{35}$S]Tic22-IV and [$^{35}$S]preSSU to immobilized POTRA-His in a dose-dependent manner (*Figure 9 C*), whereas soluble SSU was unable to compete, further demonstrating the specificity of binding. These data demonstrate the ability of Tic22-III and Tic22-IV to directly bind to Toc75 POTRA domains and are consistent with a functional interaction between these proteins in the intermembrane space.

## Discussion

The number and function of POTRA domains in Omp85 family members vary, but in all systems, they mediate interactions with transport substrates and/or factors that assist with Omp85 protein function (*Jacob-Dubuisson et al., 2009*). POTRA domains are defined by repeats of conserved $\beta_1\alpha_1\alpha_2\beta_2\beta_3$ secondary structural motifs (*Clantin et al., 2007*, *Koenig et al., 2010*, *Paila et al., 2015*), and the variation in primary structure within the repeats contributes to the functional diversification and adaptation of family members for specific roles in the transport and insertion of nascent polypeptides at the outer membrane in bacteria and mitochondria (*Sanchez-Pulido et al., 2003*, *Koenig et al., 2010*). Our analysis of Toc75 is consistent with a multifunctional cassette model for POTRA domains, and highlights the unique adaptation of the Toc75 POTRA domains to mediate

interactions that are essential for assembly of the core TOC components and the transport of polypeptides across the outer membrane.

We took advantage of the observation that expression of Toc75ΔP1, Toc75ΔP1-2 and Toc75ΔP1-3 in the heterozygous *toc75III-1* background resulted in plants with pale phenotypes and significant growth defects (*Figure 1C*). The phenotypes were dependent on the expression levels of the deletion constructs, and severe effects were observed even at levels of Toc75ΔP1, Toc75ΔP1-2 and Toc75ΔP1-3 that were far less than equimolar to endogenous native Toc75 (*Figure 1C*). Given that TOC translocons are large multimeric complexes containing component ratios estimated at 4:4:1 or 3:3:1 (Toc75:Toc34:Toc159), the insertion of defective POTRA deletion subunits into these complexes likely interferes with the coordinate functions of the native subunits.

Toc75ΔP1 and Toc75ΔP1-2 were targeted, integrated and fully processed at the outer membrane (*Figure 6A and B*), with their POTRA domains extending into the intermembrane space (*Figure 6*), which is consistent with the topology of native Toc75 and all other known Omp85 family members (*Sanchez-Pulido et al., 2003*, *Stroud et al., 2011*). This indicates that the POTRA domains do not play essential roles in targeting and membrane insertion of Toc75 itself. The orientation of the Toc75 POTRA domains in the intermembrane space also is consistent with the observation that they specifically interact with Tic22 isoforms, proteins localized to the intermembrane space (*Figure 9*). A separate study using N-terminal fusions of Toc75 to self-assembling split GFP (saGFP) concluded that the N-terminus of Toc75 was localized in the cytoplasm when transiently expressed in Arabidopsis protoplasts (*Sommer et al., 2011*). However, the saGFP-Toc75 constructs used in the study were not shown to complement the lethal phenotype of *toc75-III-1* mutants, and therefore, it is unclear if the fusions assumed a functional topology in the outer membrane.

Our studies of protein import into isolated chloroplasts from the *TOC75ΔP1#2* and *TOC75ΔP1-2#2* plants demonstrate that the POTRA deletions resulted in a significant reduction in the rate of protein import into chloroplasts (*Figure 3*). The analysis of saturation-binding experiments under low [ATP] and [GTP] revealed that the decrease in import was due to the reduction in the total number of functional import sites rather than a significant reduction in the affinity of TOC complexes for the preprotein (*Figure 5*). Furthermore, the levels of TOC components were unchanged or increased in *TOC75ΔP1#2* and *TOC75ΔP1-2#2* plants (*Figure 7*), suggesting that the decreased import capacity resulted from defective TOC function and not from decreases in the overall levels of the core TOC proteins.

Our data reveal an interesting function for the POTRA domains in TOC complex assembly that accounts, at least in part, for the decreased levels of functional TOC complexes observed in *TOC75ΔP1#2* and *TOC75ΔP1-2#2* chloroplasts. Expression (*Figure 7*) and 2D blue-native PAGE (*Figure 8*) analyses revealed significant changes in the levels and assembly of the core TOC components, which likely results in a population of defective or non-functional translocons. Toc75ΔP1 appeared to retain the ability to assemble with both Toc33 and Toc159 GTPase receptors into 1.3 MDa and 440 kDa complexes (*Kikuchi et al., 2006*), suggesting that POTRA1 is not required for stoichiometric assembly of TOC complexes. However, Toc75ΔP1-2 was only detected in novel complexes that migrated between 200–880 kDa. A major fraction of Toc33, but not native Toc75 or Toc159, exhibited a coincident broad range of distribution with Toc75ΔP1-2 (*Figure 8B*). Together with the observation that Toc75ΔP1-2 co-immunoprecipitated with Toc33, this suggests that Toc75ΔP1-2 is able to associate normally with Toc33, but not Toc159, evidenced by the lack of Toc159 in the 200–880 kDa complexes in *TOC75ΔP1-2* chloroplasts. This implicates POTRA-2 alone, or in combination with POTRA-1, in interactions between Toc75 and Toc159, which is a requirement for proper TOC complex assembly and function.

The role of Toc75 POTRAs in TOC assembly is reminiscent of the role of the five repeat POTRA domains of the BamA in the assembly of the BAM complex at the outer membrane. POTRA 5 of BamA interacts with the essential periplasmic lipoprotein, BamD, to form the core of the complex (*Malinverni et al., 2006*, *Sklar et al., 2007*, *Wu et al., 2005*). The core complex recruits BamC and E (*Albrecht and Zeth, 2011*, *Kim et al., 2007*), and POTRA 2–4 of BamA interact with BamB (*Kim et al., 2007*, *Kim and Paetzel, 2011*, *Noinaj et al., 2011*, *Dong et al., 2012*) to constitute the fully functional BAM complex. Despite the conservation of functional domains, Toc75 appears to be unique amongst Omp85 family members because it evolved to mediate protein import into plastids from the cytoplasm. The reversal of the transport activity relative to the bacterial and mitochondrial Omp85 family members occurred with the diversification of Toc75 from the cyanobacterial Omp85

ancestor and was driven by the association of Toc75 with the Toc33 and Toc159 receptors. Toc33 and Toc159 provide the GTP-dependent driving force for the unidirectional transport of preproteins from the cytoplasm into the organelle. The Toc33-Toc75-Toc159 core is a universal feature of the plant lineage, and the adaptation of POTRAs to facilitate these interactions was, therefore, a key event in the evolution of TOC complexes.

Our data also indicate that disruption of Toc75 function in *TOC75ΔP1#2* and *TOC75ΔP1-2#2* plants perturbs the known coordinate expression of the TOC components (*Sun et al. 2001*), resulting in significant over-expression of Toc33 relative to Toc75 and Toc159. The major portion of the excess Toc33 migrated at ~70 kDa on blue-native PAGE of *TOC75ΔP1#2* and *TOC75ΔP1-2#2* chloroplasts and was not associated with the other TOC components (*Figure 8B*). The excess Toc33 could contribute to the phenotypes observed in *TOC75ΔP1#2* and *TOC75ΔP1-2#2* plants if it interacted with preproteins and interfered with import. However, our quantitative binding data indicate that energy-independent and energy-dependent bindings are reduced in *TOC75ΔP1#2* and *TOC75ΔP1-2#2* plants (*Figures 4* and *5*), arguing against preprotein binding to excess Toc33 as a significant factor in *TOC75ΔP1#2* and *TOC75ΔP1-2#2* phenotypes.

We also hypothesize that the Toc75 POTRA domains play a direct role in protein translocation across the outer membrane by selectively binding to chloroplast preproteins in the intermembrane space (*Figure 9*). This proposal is consistent with our observation that the isolated POTRA domains interact with a chloroplast preprotein, preSSU, in a transit-peptide and dose-dependent manner, and previous data demonstrating an interaction between the N-terminal region of Toc75 and preproteins targeted to chloroplasts (*Ertel et al., 2005*). Several possibilities exist for the role of the POTRA-transit peptide interaction during protein import. Toc75 crosslinks to the transit peptides of preproteins during initial binding as well as during later stages of membrane translocation (*Ma et al., 1996*), suggesting that the POTRA domains could contribute to

preprotein recognition in conjunction with the TOC GTPases and/or stabilize preprotein binding to the translocon upon initial translocation through the Toc75 membrane pore. The latter possibility appears to be most likely, considering the localization of the POTRA domains in the intermembrane space, and the observation that Toc75ΔP1 or Toc75ΔP1-2 did not impact energy-independent binding of preprotein to chloroplasts (*Figures 4* and *5*).

The substrate-binding activity of POTRA domains has precedence in other Omp85 family members. The POTRA domain of BamA binds nascent β-barrel proteins as they emerge into the periplasmic space from the Sec translocon in the bacterial inner membrane, suggesting that the BamA POTRAs form a platform for the polypeptide substrate to assume the necessary secondary structure required to initiate membrane insertion (*Gatzeva-Topalova et al., 2008*, *Kim et al., 2007*, *Knowles et al., 2008*). POTRA-5, located most adjacent to the β-barrel, is proposed to play an essential role in assisting the insertion of the substrate into the membrane directly or in coordination with the β-barrel and the oligomeric BAM complex (*Sinnige et al., 2015*, *Noinaj et al., 2013*). The single POTRA of Sam50 β-barrel component of the mitochondrial SAM machinery also is proposed to assist in release of the β-barrel substrate into the mitochondrial outer membrane (*Kutik et al., 2008*, *Stroud et al., 2011*).

Recent crystal structures of BamA showed two different orientations of its POTRA domains, leading to the proposal that the domains are involved in a gating mechanism for regulating access of the substrate to the the β-barrel membrane insertase (*Noinaj et al., 2013*, *Bakelar et al., 2016*). A crystal structure and molecular dynamics simulations of POTRA domains of OMP85 from *Anabaena* revealed a similar conserved feature in these proteins (*Koenig et al., 2010*), and mutations in the unique loop region of the POTRA domains adjacent to the β-barrel in the cyanobacterial protein influenced the pore properties of the β-barrel (*Koenig et al., 2010*, *Ertel et al., 2005*). It is tempting to speculate that a similar gating mechanism might function at Toc75, thereby providing a mechanism by which transit peptide binding at TOC complexes controls the outer membrane channel by triggering reorientation of the POTRA domains.

Finally, we also demonstrate specific interactions of the Toc75 POTRA domains with two homologous import components of the intermembrane space, Tic22-III and Tic22-IV. Tic22 from pea was previously shown to covalently cross-link to preprotein import intermediates (*Kouranov et al., 1998*), and proteins with significant similarity to Tic22 from apicoplasts of *Toxoplasma gondii* and *Plasmodium falciparum* have been shown to exhibit chaperone activity (*Glaser et al., 2012*). Interestingly, a Tic22-like protein from cyanobacteria was shown to covalently cross-link with an Omp85

family member of the cyanobacterial outer membrane (*Tripp et al., 2012*), providing a precedent for the interaction of Omp85 family members with the Tic22 family of putative chaperones, and its potential conservation from the cyanobacterial endosymbiont. This interaction is also reminiscent of the interaction between the BamA POTRAs and a periplasmic chaperone, SurA, during β-barrel insertion at the outer membrane (*Bennion et al., 2010*).

The significant increase in the levels of Tic22 isoforms in *TOC75ΔP1#2* and *TOC75ΔP1-2#2* plants is consistent with a functional association between the Toc75 POTRA domains and the Tic22 chaperones and could represent a response to a disruption in chaperone-preprotein interactions during membrane translocation. To date, a chaperone activity for Tic22-III and Tic22-IV in plants has not been investigated, and it will be of great interest to understand how the Toc75 POTRA domains might coordinate interactions between preproteins and the putative chaperone to facilitate preprotein transit through the intermembrane space and engagement of the TIC complex.

Our analysis of Toc75 is consistent with models in which the POTRA domains play versatile roles in TOC assembly and function, which account for the decreased levels of functional TOC complexes seen in *TOC75ΔP1#2* and *TOC75ΔP1-2#2* chloroplasts. As is the case in other Omp85 family members, each domain is likely to contribute individual activities, but also work in association with the other domains to mediate assembly of the translocon complex and coordinate preprotein translocation. In contrast to bacterial and mitochondrial outer membranes, chloroplasts contain two Omp85 family members: Toc75 and OEP80 (Outer Envelope Protein 80) (*Hsu and Inoue, 2009*). Similar to Toc75, OEP80 is essential in Arabidopsis (*Patel et al., 2008*) and is predicted to contain three N-terminal soluble POTRA domains and a C-terminal β-barrel (*Hsu and Inoue, 2009*, *Hsu et al., 2012*). OEP80 is not detected in isolated oligomeric TOC complexes, and it has been proposed to function in β-barrel protein biogenesis in a manner analogous to Sam50 and BamA (*Eckart et al., 2002*). Although phylogenetic analyses indicated that Toc75 and OEP80 are more closely related to each other than either is to other bacterial Omp85 family members, distinct sequence signatures have been identified in both the POTRA and β-barrel domains of the chloroplast proteins to account for evolutionary divergence of these two β-barrel channels (*Day et al., 2014*). The evolution of an import apparatus to mediate the transport of nucleus-encoded preproteins into plastids was an essential element of endosymbiosis. Our data are consistent with the divergent evolution of Toc75 and OEP80 from the ancestral cyanobacterial *OMP85* gene to retain the pathway for β-barrel protein biogenesis, while providing a novel pathway for protein import into the organelle.

## Materials and methods

### Plant growth, genetic complementation of *toc75-III-1* and screening of transgenic plants

Seedlings of *Arabidopsis thaliana* Col-0 were grown on phytoagar plates containing 0.5 × Murashige and Skoog growth medium (MS medium), 1% sucrose under long-day condition for 14-day at 22℃. Genetic complementation was carried out as described earlier (*Shipman-Roston et al., 2010*) with a few modifications. Briefly, the Arabidopsis *TOC75* genomic fragment of 4.7 kb, including 1 kb upstream and 0.5 kb downstream of the coding sequence in pDONR221 was generated as described (*Shipman-Roston et al., 2010*), producing the entry clone pDONRgTOC75wt. All deletion constructs were made using the Quickchange II XL site-directed mutagenesis kit (*Makarova et al., 2000*) using the primers listed in *Table 2*. The expression clones were generated by an LR recombination reaction between entry clones and the Gateway destination vector, pBnRGW (http://gateway.psb.ugent.be). The destination vector contains a fluorescent marker, DsRed, under a seed specific promoter and BASTA (glufosinate ammonium) resistance, allowing for rapid screening of seed and seedlings. Final constructs were confirmed by sequencing and introduced into heterozygous *toc75-III-1* plants (*Baldwin et al., 2005*) by the *Agrobacterium tumefaciens*-mediated floral dip method (*Clough and Bent, 1998*). Transformed seed was initially screened for DsRed fluorescence, followed by growth on MS plates containing 20 μg/ml hygromycin to select for plants carrying the *TOC75* variants and *toc75-III-1*, respectively.

## Reverse transcription-PCR and quantitative PCR

Total RNA was isolated from 14-day-old Arabidopsis seedlings grown on plates using the RNeasy Plant Mini Kit (Qiagen, Germany) as per manufacturer's instructions. RNA isolated was quantified by absorbance at 260 nm, and equivalent amounts were used to synthesize first-strand cDNA using SuperScript III and an oligo(dT) primer (Invitrogen, Carlsbad, CA). cDNA was used for PCR reactions (35 cycles) using gene-specific primers. Primers used for PCR amplification of *TOC75, TOC75ΔP1, TOC75ΔP1-2* or *TOC75ΔP1-3* are listed in *Table 3*.

Primers for RT-qPCR reactions were designed by Primer 3 v.0.4.0 (*Untergasser et al., 2012*) with optimal melting temperatures of 58–60°C, primer lengths of 20–30 bp and amplicon lengths of 220–250 bp. RT-qPCR was performed in triplicate using DyNAmo Flash SYBR Green qPCR kit (Thermo Scientific, Waltham, MA). Baseline and threshold cycles (Ct) were analyzed by Realplex 2.2 Software. Relative gene expressions were calculated with respect to internal reference of actin using the $2^{\Delta}Ct$ method (*Schmittgen and Livak, 2008*).

## Chloroplast isolation

Intact chloroplasts were isolated from 14-day-old plants as described previously (*Schulz et al., 2004*, *Wang et al., 2008*, *Brock et al., 1993*). Chloroplasts were counted with a hemocytometer using a phase-contrast microscope with 40x objective (*Sung and Chen, 1989*) and equivalent numbers of chloroplasts were used in all assays. Chloroplasts were fractionated into soluble and membrane fractions by centrifugation at 18,000 x g at 4°C for 30 min. Proteins from the soluble fraction were precipitated with 15% trichloroacetic acid. Both fractions were resuspended in the sample buffer and analyzed by SDS-PAGE followed by immunoblotting with anti-atToc75.

## Protease treatments

Thermolysin and trypsin treatments of intact chloroplasts were performed as described earlier (*Pain and Blobel, 1987*, *Chen and Schnell, 1997*, *Inoue et al., 2013*, *Hsu et al., 2012*). All protease treatments were performed in HS buffer (50 mM Hepes-KOH, 330 mM sorbitol, pH 7.5) for 30 min on ice with increasing protease concentrations up to 200 µg/ml trypsin and 100 µg/ml thermolysin in a reaction containing $5 \times 10^8$ chloroplasts/ml. Thermolysin was quenched with 20 mM EDTA and trypsin digestion was quenched by addition of protease inhibitor cocktail (2.5 mM PMSF, 0.05 mg/ml Nα-Tosyl-L-lysine chloromethyl ketone (TLCK), 0.25 mg/ml soybean trypsin inhibitor and 2 µg/ml aprotinin). Intact chloroplasts were reisolated through 35% Percoll and washed with ice-cold HS buffer containing EDTA or trypsin inhibitor cocktail. 1% (v/v) Triton X-100 was included in reactions to disrupt membrane permeability.

## Protein extraction, immunoblotting and immunoaffinity chromatography

Total protein was extracted directly in SDS-PAGE sample buffer from Arabidopsis seedlings. Samples corresponding to equivalent amounts of total protein were resolved by SDS-PAGE, transferred to PVDF membranes, and subjected to immunoblotting with antisera to the indicated proteins. Immunoblotting was performed as described previously (*Ma et al., 1996*) using chemiluminescence detection. Antisera to atToc159, atToc132, atToc33, atToc34, atTic110, LHCP, SSU and OEP80 were described previously (*Ivanova et al., 2004*, *Inaba et al., 2005*, *Wang et al., 2008*, *Hsu et al., 2012*). atToc75 antiserum was a generous gift of Dr. Takehito Inaba at Miyazaki University. Quantitation of immunoblots was carried out using ImageJ (v. 1.47).

Immunoaffinity chromatography of TOC core complex proteins under native conditions was performed as described previously (*Kouranov et al., 1998*). The proteins in each fraction were precipitated with 10% trichloroacetic acid. Total membranes, unbound and eluate fractions were analyzed by SDS-PAGE and immunoblotting with antiserum of atToc159, atTic110, atToc75 and affinity purified atToc33 antibodies.

## In vitro translation and preprotein binding and import assays

[$^{35}$S]Methionine-labeled Arabidopsis preSSU, preE1α, fully processed (mature) forms of Toc75ΔP1 and Toc75ΔP1-2 were generated in a coupled transcription-translation system containing reticulocyte lysate according to the manufacturer's instructions (Promega, Madison, WI). The in vitro

translation product containing [$^{35}$S]preSSU or [$^{35}$S]preE1$\alpha$ was used directly for chloroplast import assays. Chloroplast early binding and import reactions were performed using [$^{35}$S]methione-labeled preproteins and equal number of chloroplasts (10$^7$ chloroplasts), in a total volume of 50 µl of import buffer (330 mM sorbitol, 50 mM Hepes-KOH, pH 7.5, 25 mM KOAc and 5 mM MgOAc) for 5 min with 0.1mM ATP and GTP or 20 min at 26℃ in the presence of 3 mM ATP for binding and import, respectively as described previously (*Chen et al., 2002*, *Wang et al., 2008*). For preprotein binding reactions or early import intermediate formation, the in vitro translated [$^{35}$S]preSSU was gel filtered on Sephadex G-25 to remove nucleotides (*Agne et al., 2009*). For the chase experiments, chloroplasts (10$^7$) containing bound preprotein or the early import intermediate reactions were generated as above. Chloroplasts recovered after isolation over the 35% Percoll cushion were washed once and resuspended in import buffer. Preprotein translocation was initiated with the addition of 3 mM ATP, and samples were incubated at 26℃ for 20 min. All samples were resolved by SDS-PAGE and analyzed by phosphorimaging (Fuji Fla-5000 phosphorimager). Equivalent numbers of chloroplasts based on microscopic counting were loaded in all lanes. ImageQuant TL (v 1.00) software was used for analysis.

For saturation-binding experiments, preSSU-FLAG-HIS was expressed from pET21a:atpSC(-1)-3xFLAG-HIS in *E. coli* BL21 (DE3) and purified using Ni-NTA matrix (Novagen) under denaturing conditions as described previously (*Smith et al., 2004*, *Inoue et al., 2013*). Import and binding were performed as previously described (*Wang et al., 2008*). The preSSU-FLAG-HIS was denatured with 6 M urea and diluted into import buffer, before the addition of 10$^7$ chloroplasts to initiate the reaction. The maximum number of binding sites and apparent $K_d$ for early binding were calculated by plotting specific binding vs. [S], where specific binding was quantitated from immunoblots using ImageJ and [S] equaled the total concentration of preSSU-FLAG-HIS, using non-linear regression analysis of binding data using Graphpad Prism software version 4.00 (San Diego, CA).

## 2D Blue-native PAGE analysis

2D BN-PAGE was performed as described earlier (*Kikuchi et al., 2006*, *Hsu et al., 2012*) with a few modifications. Briefly, intact chloroplasts isolated from *A. thaliana* were resuspended in the hypotonic lysis buffer (10 mM HEPES-KOH, pH 7.5, containing 1 mM MgCl$_2$ and 10 µl/ml PIC [Protease Inhibitor Cocktail for plant extracts; P-9599, Sigma-Aldrich, St. Louis, MO]) for 10 min on ice, and centrifuged at 18,000 x g, 4℃ for 30 min. The pellet was solubilized in the buffer (50 mM Bis(2-hydroxyethyl) iminotris (hydroxymethyl)methane (BisTris)-HCl, pH 7.0, containing 1% (w/v) decylmaltoside (Calbiochem, San Diego, CA), 500 mM 6-amino-n-caproic acid, 10% (v/v) glycerol, and 10 µl/ml PIC) at a concentration of 0.5 mg chlorophyll/ml for 20 min on ice. The insoluble materials were removed by centrifugation at 17,000 g for 20 min at 4℃. The supernatant was mixed with Coomassie Brilliant Blue G-250 solution (50 mM BisTris-HCl, pH 7.0, containing 5% (w/v) Serva blue G and 500 mM 6-amino-n-caproic acid] in 100:3.25 (v/v) to give a ratio of the detergent (decylmaltoside) to the dye (Serva blue G) as 8:1 (w/w). The sample was loaded onto a 4–12% (w/v) polyacrylamide gradient gel in 50 mM BisTris-HCl, pH 7.0, containing 500 mM 6-amino-n-caproic acid (1.5 mm thickness, Mini-PROTEAN3, Bio-Rad laboratories, Hercules, CA). Electrophoresis was carried out in the cathode buffer (50 mM tricine, 15 mM BisTris-KOH, pH 7) and the anode buffer (50 mM BisTris-HCl, pH 7) at a constant voltage of 30 V at 4℃ for 14 hr. Spectra multicolor Broad Range Protein Ladder (Thermo Scientific) and ferritin (Sigma, F-4503) were used as size standards. After BN-PAGE, the marker lanes were directly stained with Coomassie Brilliant Blue, and the sample lanes were excised and heated at 37℃ for 30 min in 3.3% SDS, 4% 2-mercaptoethanol and 65 mM Tris-HCl, pH 6.8. The gel strip was layered on top of a 1.5 mm-thick stacking gel, and the second dimension 5–12% gradient SDS-PAGE under reducing conditions was performed according to standard procedures.

## Solid phase binding assays

A glutathione S-transferase-POTRA fusion (GST-POTRA-His) was expressed in *E. coli*, purified by Glutathione Sepharose 4B chromatography and cleaved using TEV protease to generate soluble POTRA-His. Solid phase binding assays were performed as described previously (*Smith et al., 2004*) with some modifications. Soluble POTRA-His was bound to ~10 µl of packed Ni-NTA resin and washed thrice with 50 mM Hepes-KOH, pH 7.5, 2 mM MgCl$_2$, and 40 mM KOAc (HMK buffer) with 10 mM imidazole and 0.1% Triton X-100 (binding buffer). 1–3 µl of [$^{35}$S]Tic22-III, [$^{35}$S]Tic22-IV, [$^{35}$S]

preSSU and [$^{35}$S]SSU were incubated in binding buffer with POTRA-bound resin in a final volume of 100 μl for 30 min at room temperature (23°C). After washing, resin-bound proteins were eluted with SDS-PAGE sample buffer containing 500 mM imidazole. All radiolabeled proteins from in vitro pull-down assays were resolved using SDS-PAGE, and radioactivity was analyzed by phosphorimaging (Fuji Fla-5000 phosphorimager) and quantitated using ImageQuant TL (v 1.00).

## Accession numbers

Sequence data from this article can be found in the EMBL/GenBank data libraries under accession number Arabidopsis AGI locus identifier: AT3G46740 for Toc75-III.

## Acknowledgements

This work was supported by National Institutes of Health Grant 2RO1-GM061893 to DJS. and US National Science Foundation Grant no.1050602 from the Division of Molecular and Cellular Biosciences to KI LGLR is a recipient of a Postdoctoral Fellowship from the Natural Sciences and Engineering Research Council of Canada. We would like to thank Joshua Endow for providing expert technical assistance, and Dr. Takehito Inaba for providing the atToc75 antisera.

## Additional information

### Funding

| Funder | Grant reference number | Author |
|---|---|---|
| National Institute of General Medical Sciences | 2RO1-GM061893 | Danny J Schnell |
| National Science Foundation | MCB 1050602 | Kentaro Inoue |
| Natural Sciences and Engineering Research Council of Canada | Postdoctoral Fellowship | Lynn GL Richardson |

The funders had no role in study design, data collection and interpretation, or the decision to submit the work for publication.

### Author contributions

YDP, Acquisition of data, Analysis and interpretation of data, Drafting or revising the article; LGLR, This author was added because the first author of this manuscript (Paila) took parental leave with her first child right after we received the comments from reviewers. Dr. Richardson completed the experimental and textual revisions of the manuscript. Conception and design, Acquisition of data, Analysis and interpretation of data, Drafting or revising the article; HI, Conception and design, Acquisition of data, Contributed unpublished essential data or reagents; ESP, JM, Acquisition of data, Analysis and interpretation of data, Contributed unpublished essential data or reagents; KI, Conception and design, Drafting or revising the article, Contributed unpublished essential data or reagents; DJS, Conception and design, Analysis and interpretation of data, Drafting or revising the article

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
