## [Decision Letter]

Thank you for submitting your work entitled "Multi-functional roles for the polypeptide transport associated (POTRA) domains of Toc75 in chloroplast protein import" for consideration by *eLife*. Your article has been reviewed by three peer reviewers, and the evaluation has been overseen by Nikolaus Pfanner as the Reviewing Editor and Vivek Malhotra as the Senior Editor.

The reviewers have discussed the reviews with one another and the Reviewing Editor has drafted this decision to help you prepare a revised submission.

Summary:

The chloroplast protein Toc75 is a member of the Omp85 family of proteins. Proteins of this family contain a β-barrel membrane-embedded domain in their C-terminal part and one to five POTRA domains in the N-terminal region. However, little is known about the functions of the *TOC75* POTRA domains and there is also uncertainty about the orientation of these domains in the outer membrane. The manuscript describes a first attempt at a comprehensive analysis of the *TOC75* POTRA domains and also addresses the membrane orientation. The authors analyzed the outcome of deleting one, two, or all three POTRA domains of Toc75. In addition they characterized the import behavior of organelles coexpressing one of the deletion variants together with a native form. They also analyzed the assembly of the TOC complex in such chloroplasts. An interesting side aspect of this report is the Toc75 topology analysis, which is in agreement with the bacterial and mitochondrial Omp85 homologs. This is important since a previous controversial study (Sommer et al., 2011) proposed just the opposite orientation for chloroplasts. Finally, the authors show that a recombinant polypeptide corresponding to the three POTRA domains can interact with precursor protein and with Tic22. In general, the data presented in the manuscript is technically sound and represents the first characterization of the Toc75 POTRA domains for protein import into chloroplasts. The elucidation of *TOC75* POTRA function will represent a significant step forward in the field, and so this manuscript is potentially of considerable interest.

Essential revisions:

1) Three *TOC75* POTRA deletion constructs were used to try and complement the *toc75-III-1* mutant. These were dP1, dP1-2, dP1-3. As none of these was able to complement the mutant, this only reveals a necessity for POTRA 1. Alternatively perhaps the folding of the whole POTRA domain was already disrupted by the dP1 deletion and so further deletions had no further effects.

Putative dominant-negative phenotypes were found in some transformed plants containing the POTRA deletion constructs in the heterozygous *toc75-III-1* background. Single plant photos show an increasing severity of effect related to deletion of the P1, P1&2 or P1, 2&3 POTRA domains. Transformants with no phenotype are shown to have near WT levels of overexpression, whereas it appears that the negative phenotypes are associated with high overexpression of the deletion constructs. However, as only one WT-like and one dominant-negative line containing each construct is assessed, the asserted dosage dependency is not clear (this is referring to Figure 1 and Figure 2). Adding some lines of intermediate expression and with intermediate phenotypes would help. Related to this it would be useful if it could be more clearly shown that the phenotypes in Figure 1 are truly representative and characteristic of each construct.

2) There is a discrepancy between the results of Figure 2 and those of Figure 2. Whereas the mRNA levels of *TOC75ΔP1#2* and *Toc75ΔP1-2#2* are much higher than those of *TOC75* (Figure 2) the protein levels of these variants are much lower than those of Toc75. The authors do not comment on this point.

According to Figure 2, Figure 5, and Figure 7 the levels of *Toc75ΔP1-2#2* are at most about 20% of those of native Toc75. It is not clear how such low levels can cause such strong "dominant negative" effects. Can the authors exclude the possibility that part of these variant molecules are clogging the TOC complex as substrate in transit and thus affecting function and assembly of the TOC complex?

The phenotypes in Figure 1 and Figure 2 are interpreted as a competition effect between the POTRA-deleted *TOC75* and WT *TOC75*, implying necessity of the POTRA domains (this is supported by data in Figure 6 showing that POTRA-deleted *TOC75* was imported in vitro and has the same in vivo topology as *TOC75*). However results for a highly overexpressed *TOC75* line are not shown. Therefore it is not possible to tell if the deleterious effect is truly due to assembled *TOC75* lacking POTRA domains (rather than e.g. overabundance of the protein causing competition for import in vivo, presence of the β-barrel domain, or excess import intermediate in the intermembrane space in vivo).

The title asserts multifunctional roles for POTRA domains in TOC75. However many of the analyses (Figure 3, Figure 4, Figure 5, Figure 7) of the dominant negative dP1 and dP1-2 plants only show a clear difference compared to the WT-like *TOC75* and *toc75-III-1+/-*, but not between each other. The only data potentially revealing differing roles of the POTRAs are in Figure 1 and Figure 8, which show that dP1-2 *TOC75* does not form part of the largest TOC complex.

3) Figure 3, Figure 4 and Figure 5: The conclusions regarding the hampered import capacity of organelles with the deletion variants are based on only ONE precursor protein (preSSU) and thus might reflect an exceptional case. To substantiate the claims the authors should check at least one more client protein.

As shown in Figure 7, the Toc33 protein level are strongly increased in *TOC75ΔP1#2* and *Toc75ΔP1-2#2* chloroplasts while Toc159 remains unchanged. Additionally, Figure 8 nicely shows that Toc33 accumulates in the outer envelope without being assembled in the mature TOC complex. Thus, a major concern of the study is the question if the reduced import of preSSU observed for *TOC75ΔP1#2* and *TOC75ΔP1-2#2* in Figure 3 and Figure 4 is simply caused by preprotein binding to unassembled Toc33 which cannot hand the substrate over to Toc75? In this case, the initial preprotein binding would not be affected but the import across the outer envelope would be impaired as observed in Figure 4. This could be tested e.g. by overexpression of Toc33 in the background of wildtype Toc75 or by testing precursors which do not depend on Toc33.

4) Figure 6: Important parts of the anti-myc Western in 6D (iii), which should show the full-length Toc75, were unfortunately omitted. These parts should be shown. In contrast to the authors´ claims, it seems that *Toc75ΔP1_myc_* is behaving differently than Toc75 as the former is completely resistant to trypsin. Surprisingly, in the anti-POTRA western is seems as if the vast majority of molecules of both forms are resistant to trypsin. Since the authors have an antibody against Toc75 it will be informative to show a full gel immunodecorated with anti-Toc75 to see also smaller fragments. Currently, I am not convinced that *TOC75ΔP1* has the same topology as native Toc75.

5) In Figure 8 the authors performed an anti-atToc33 co-immunoprecipitaion to analyze the stoichiometric assembly of the TOC complex. However, due to the low expression levels of *TOC75ΔP1* and *TOC75ΔP1-2* this analysis does not allow any conclusion about the interaction between *TOC75ΔP1* and *TOC75ΔP1-2* and the other Toc and Tic subunits. Alternatively, the authors could perform an anti-myc co-immunoprecipitation using their Toc75_myc_ chloroplasts.

Figure 8 lacks a negative control. As it stands it might be that solubilization was not complete and ALL outer membrane proteins are pulled-down with Toc33. A negative control like Oep80 or other outer membrane protein should be included.

Figure 8 raises several questions: (a) Kikuchi et al. reported two species of TOC complex, one at 800-1000 kDa and the other at 350-500 kDa. Why are the results in Figure 8 different? (b) Why is in part (iii) the ratio of the intensities of the bands representing full-length Toc75 and *Toc75ΔP1-2#2* very similar, whereas in many other experiments (like Figure 7, Figure 8 and others) the variant is detected in much lower levels? (c) The Western of Toc159 should include also the range of 86 kDa to estimate how much of the molecules were cleaved. Do the complexes in the 400-800 kDa range contain the cleaved version? (d) in panel (ii): why are there three species of Toc75? A figure with a better quality and more distinct bands is required to draw conclusions about the assembly of the TOC complex.

---

## [Author Response]

*Essential revisions:*

1) Three TOC75 POTRA deletion constructs were used to try and complement the toc75-III-1 mutant. These were dP1, dP1-2, dP1-3. As none of these was able to complement the mutant, this only reveals a necessity for POTRA 1. Alternatively perhaps the folding of the whole POTRA domain was already disrupted by the dP1 deletion and so further deletions had no further effects.

*Putative dominant-negative phenotypes were found in some transformed plants containing the POTRA deletion constructs in the heterozygous toc75-III-1 background. Single plant photos show an increasing severity of effect related to deletion of the P1, P1&2 or P1,2&3 POTRA domains. Transformants with no phenotype are shown to have near WT levels of overexpression, whereas it appears that the negative phenotypes are associated with high overexpression of the deletion constructs. However, as only one WT-like and one dominant-negative line containing each construct is assessed, the asserted dosage dependency is not clear (this is referring to Figure 1 and Figure 2). Adding some lines of intermediate expression and with intermediate phenotypes would help. Related to this it would be useful if it could be more clearly shown that the phenotypes in Figure 1 are truly representative and characteristic of each construct.*

a) The reviewers asked us to clarify our conclusion that the three POTRA domains are essential for Toc75 function. We agree with the reviewers that we can only conclude from our analyses that POTRA1 is essential for Toc75 function. We have modified the Abstract to avoid implying otherwise. However, the increasing severity of the lines expressing *TOC75ΔP1, TOC75ΔP1-2*, and *TOC75ΔP1-3* is consistent with our conclusion that P2 and P3 also contribute to the function of Toc75. This also is supported by the data in Figure 8, which show that TOC complexes from *TOC75ΔP1#2* and *TOC75ΔP1-2#2* have distinct mobility patterns on 2D blue-native PAGE. This leads us to hypothesize that the deletion of POTRA1 and 2 impacts TOC assembly in a manner that is not observed when only POTRA1 is absent.

b) Second, the reviewers raised the issue of whether our analyses of *TOC75ΔP1#2* and *TOC75ΔP1-2#2* are representative of multiple lines expressing the POTRA deletion constructs. We have included data on the phenotypes and levels of expression of the POTRA deletions from six independent transformed lines each for *TOC75ΔP1* and *TOC75ΔP1-2* (Figure 1—figure supplement 1 and Figure 2—figure supplement 1). These lines provide evidence that our data are representative of the constructs and support our conclusion that the severity of the phenotypes correlates with the levels of *Δ*P1 and *Δ*P1-2 expression. It is very difficult to quantify the correlation between expression and visible phenotype in transgenic plants. Therefore, we have eliminated the term “dose-dependent” from the text to avoid implying that we have performed a classical dose-response analysis. Instead, we simply state that increased expression of the deletion mutants results in more severe phenotypes (paragraph six, Introduction).

*2) There is a discrepancy between the results of Figure 2 and those of Figure 2. Whereas the mRNA levels of TOC75ΔP1#2 and Toc75ΔP1-2#2 are much higher than those of TOC75 (Figure 2) the protein levels of these variants are much lower than those of Toc75. The authors do not comment on this point.*

*According to Figure 2, Figure 5, and Figure 7 the levels of Toc75ΔP1-2#2 are at most about 20% of those of native Toc75. It is not clear how such low levels can cause such strong "dominant negative" effects. Can the authors exclude the possibility that part of these variant molecules are clogging the TOC complex as substrate in transit and thus affecting function and assembly of the TOC complex?*

*The phenotypes in Figure 1 and Figure 2 are interpreted as a competition effect between the POTRA-deleted TOC75 and WT TOC75, implying necessity of the POTRA domains (this is supported by data in Figure 6 showing that POTRA-deleted TOC75 was imported in vitro and has the same in vivo topology as TOC75). However results for a highly overexpressed TOC75 line are not shown. Therefore it is not possible to tell if the deleterious effect is truly due to assembled TOC75 lacking POTRA domains (rather than e.g. overabundance of the protein causing competition for import in vivo, presence of the β-barrel domain, or excess import intermediate in the intermembrane space in vivo).*

*The title asserts multifunctional roles for POTRA domains in TOC75. However many of the analyses (Figure 3, Figure 4, Figure 5, Figure 7) of the dominant negative dP1 and dP1-2 plants only show a clear difference compared to the WT-like TOC75 and toc75-III-1+/-, but not between each other. The only data potentially revealing differing roles of the POTRAs are in Figure 1 and Figure 8, which show that dP1-2 TOC75 does not form part of the largest TOC complex.*

These comments focus on the expression levels of the Toc75 POTRA deletions and wild type Toc75 in transgenic plants.

a) We did not examine the cause of the high levels of mRNA accumulation of the POTRA deletions in the *TOC75ΔP1#2* and *TOC75ΔP1-2#2* lines relative to expression of wild type Toc75. The transgenes are all expressed from the native promoter, and the gene constructs retain all corresponding intron and exon organization. However, it is well known that positional effects due to the sites of insertion in the *Arabidopsis* genome can result in significant differences in the levels of expression of the same gene constructs. With regards to the fact that the protein levels of the POTRA deletions are significantly lower than wild type Toc75 protein, we speculate that this might be due to relative instability of the deletion constructs and their apparent toxicity to the plants. Statements to these effects are added to the text (paragraph three, subheading “POTRA domains are required for Toc75 function”).

b) TOC complexes are large multimeric assemblies. Toc75 appears to self-associate when expressed and reconstituted from *E. coli*. We hypothesize that the introduction of defective subunits into these complexes can disrupt the function of the large complexes by interfering with the coordinate activities of Toc33, Toc159 and Toc75. This hypothesis is consistent with other known cases in which mutations that impact a small proportion of subunits of oligomeric complexes have significant impacts on function. We have added a statement to the text to explain this possibility (paragraph one, Discussion).

c) Our data do not support the possibility that the POTRA deletions might be clogging the translocon channel and thereby causing protein import defects. The deletions are targeted, fully processed and membrane integrated as shown in Figure 6. Furthermore, protein import intermediates that are stuck in the TOC translocon are predicted to be sensitive to thermolysin treatment in intact chloroplasts. The POTRA deletions are insensitive to thermolysin and, in the case of *TOC75ΔP1*, assume the same topology as native Toc75 (Figure 6).

d) It is very difficult to obtain highly over-expressed lines of wild type Toc75. High level overexpression leads to co-suppression and a lethal phenotype. In unpublished work, we previously expressed Toc75 under strong constitutive promoters (e.g. 35S CaMV), and could obtain up to a 2-fold increase in expression similar to what we observe for wild type Toc75 in *TOC75ΔP1#2* without any obvious phenotypes.

e) As stated above, we agree that our data do not demonstrate that the individual POTRA repeats have entirely distinct functions. However, it is clear that the POTRAs together play multiple roles in TOC function and assembly. Furthermore, the fact that *TOC75ΔP1* and *TOC75ΔP1-2* have distinct effects on TOC assembly (Figure 8) suggest that their impact on import complexes is not equivalent.

*3) Figure 3, Figure 4 and Figure 5: The conclusions regarding the hampered import capacity of organelles with the deletion variants are based on only ONE precursor protein (preSSU) and thus might reflect an exceptional case. To substantiate the claims the authors should check at least one more client protein.*

*As shown in Figure 7, the Toc33 protein level are strongly increased in Toc75ΔP1#2 and Toc75ΔP1-2#2 chloroplasts while Toc159 remains unchanged. Additionally, Figure 8 nicely shows that Toc33 accumulates in the outer envelope without being assembled in the mature TOC complex. Thus, a major concern of the study is the question if the reduced import of preSSU observed for TOC75ΔP1#2 and TOC75ΔP1-2#2 in Figure 3 and Figure 4 is simply caused by preprotein binding to unassembled Toc33 which cannot hand the substrate over to Toc75? In this case, the initial preprotein binding would not be affected but the import across the outer envelope would be impaired as observed in Figure 4. This could be tested e.g. by overexpression of Toc33 in the background of wildtype Toc75 or by testing precursors which do not depend on Toc33.*

The reviewers raised questions regarding the nature of the protein import defect in the POTRA deletion lines.

a) We have added import experiments with an additional preprotein, the precursor to the E1 α subunit of pyruvate dehydrogenase (Figure 3). This precursor has previously been shown to utilize TOC complexes for import that are distinct from those used by preSSU. These complexes contain Toc75 assembled with different isoforms of the TOC GTPases. The data show that import of preE1 α is reduced to the same degree as preSSU import and support our conclusion that the POTRAs affect import generally (paragraph three, subheading “POTRA domains are required for Toc75 function”).

b) Although we cannot eliminate the possibility that excess Toc33 contributes to the import defect, we would expect to see increased binding to chloroplasts under no energy conditions (Figure 4). Rather, we observe slightly decreased binding levels. Similarly, the kinetic analysis in Figure 5 does not indicate significant changes in the K_d(app)_. We would expect that significant binding by “free” Toc33 would impact the K_d(app)_ as this interaction would most certainly have a distinct affinity from intact TOC complexes.

*4) Figure 6: Important parts of the anti-myc Western in 6D (iii), which should show the full-length Toc75, were unfortunately omitted. These parts should be shown. In contrast to the authors´ claims, it seems that Toc75ΔP1_myc_ is behaving differently than Toc75 as the former is completely resistant to trypsin. Surprisingly, in the anti-POTRA western is seems as if the vast majority of molecules of both forms are resistant to trypsin. Since the authors have an antibody against Toc75 it will be informative to show a full gel immunodecorated with anti-Toc75 to see also smaller fragments. Currently, I am not convinced that Toc75ΔP1 has the same topology as native Toc75.*

We believe that the questions regarding the interpretation of results in Figure 6 were a consequence of low quality images generated during PDF conversion. We apologize to the reviewers. We have replaced the original overexposed images in Figure 6 with higher quality images. We believe these data now clearly demonstrate that native Toc75 and *TOC75ΔP1* exhibit indistinguishable protease sensitivity patterns, consistent with our conclusion that they assume the same topology in the outer membrane.

*5) In Figure 8 the authors performed an anti-atToc33 co-immunoprecipitaion to analyze the stoichiometric assembly of the TOC complex. However, due to the low expression levels of Toc75ΔP1 and Toc75ΔP1-2 this analysis does not allow any conclusion about the interaction between Toc75ΔP1 and Toc75ΔP1-2 and the other Toc and Tic subunits. Alternatively, the authors could perform an anti-myc co-immunoprecipitation using their Toc75_myc_ chloroplasts. Figure 8 lacks a negative control. As it stands it might be that solubilization was not complete and ALL outer membrane proteins are pulled-down with Toc33. A negative control like Oep80 or other outer membrane protein should be included.*

*Figure 8 raises several questions: (a) Kikuchi et al. reported two species of TOC complex, one at 800-1000 kDa and the other at 350-500 kDa. Why are the results in Figure 8 different? (b) Why is in part (iii) the ratio of the intensities of the bands representing full-length Toc75 and Toc75ΔP1-2#2 very similar, whereas in many other experiments (like Figure 7, Figure 8 and others) the variant is detected in much lower levels? (c) The Western of Toc159 should include also the range of 86 kDa to estimate how much of the molecules were cleaved. Do the complexes in the 400-800 kDa range contain the cleaved version? (d) in panel (ii): why are there three species of Toc75? A figure with a better quality and more distinct bands is required to draw conclusions about the assembly of the TOC complex.*

These comments address the assembly of the POTRA deletions with other TOC components.

a) The intent of the data in Figure 8 was only to establish that the deletion mutants can interact directly or indirectly with TOC components and was not intended to address the issue of stoichiometry. We have added a statement to the text to clarify the intent (paragraph two, subheading “Protein import components accumulate to higher levels in plants expressing POTRA deletions”).

b) We attempted to use anti-myc to immunoprecipitate complexes, but were unsuccessful, perhaps due to the inaccessibility of the epitope on Toc75 within TOC assemblies.

c) We also included immunoblots of OEP80, an outer membrane negative control, in Figure 8 to show that it does not co-immunoprecipitate with anti- Toc33.

d) The minor size differences in the TOC complexes shown here compared to the previous study is likely due to the fact that we are analyzing *Arabidopsis* chloroplasts, whereas the published study was analyzing pea chloroplasts. The GTPase receptors are different sizes in the two organisms. We do not think this is significant. To avoid confusion, we now refer to the complexes we observe as 1.3 MDa and 440 kDa complexes and explain the slight differences from the results of Kikuchi et al. in the text (paragraph one, subheading “POTRA deletions disrupt TOC complex stoichiometry”).

e) We have repeated the 2D blue-native gel analysis and included higher quality images in Figure 8 that are more representative of the ratios of Toc75 and *TOC75ΔP1* observed in other experiments.

f) We did not detect an 86 kDa fragment when the blots were probed with an anti-serum to the C-terminal domain of Toc159. We take extra precautions to avoid proteolysis during our experiments. Furthermore, Toc159 is much less susceptible to proteolysis in *Arabidopsis* than in pea chloroplasts in our hands.

g) As mentioned above, we repeated the 2D blue-native gel analysis and included higher quality blots in Figure 8. A third background band is not detected in these experiments.